# FedProf: Selective Federated Learning with Representation Profiling

## Abstract

Federated Learning (FL) has shown great potential as a privacy-preserving solution to learning from decentralized data that are only accessible to end devices (i.e., clients). In many scenarios however, a large proportion of the clients are probably in possession of low-quality data that are biased, noisy or even irrelevant. As a result, they could significantly slow down the convergence of the global model we aim to build and also compromise its quality. In light of this, we propose FedProf, a novel algorithm for optimizing FL under such circumstances without breaching data privacy. The key of our approach is a data representation profiling and matching scheme that uses the global model to dynamically profile data representations and allows for low-cost, lightweight representation matching. Based on the scheme we adaptively score each client and adjust its participation probability so as to mitigate the impact of low-value clients on the training process. We have conducted extensive experiments on public datasets using various FL settings. The results show that FedProf effectively reduces the number of communication rounds and overall time (up to 4.5x speedup) for the global model to converge and provides accuracy gain.

## 1 Introduction

With the advances in Artificial Intelligence (AI), we are seeing a rapid growth in the number of AI-driven applications as well as the volume of data required to train them. However, a large proportion of data used for machine learning are often generated outside the data centers by distributed resources such as mobile phones and IoT (Internet of Things) devices. It is predicted that the data generated by IoT devices will account for 75% of the total in 2025 (Meulen, 2018). Under this circumstance, it will be very costly to gather all the data for centralized training. More importantly, moving the data out of their local devices (e.g., mobile phones) is now restricted by law in many countries, such as the General Data Protection Regulation (GDPR)[1] enforced in EU.

We face three main difficulties to learn from decentralized data: i) massive scale of end devices; ii) limited communication bandwidth at the network edge; and iii) uncertain data distribution and data quality. As an promising solution, Federated Learning (FL) (McMahan et al., 2017) is a framework for efficient distributed machine learning with privacy protection (i.e., no data exchange). A typical process of FL is organized in rounds where the devices (clients) download the global model from the server, perform local training on their data and then upload their updated local models to the server for aggregation. Compared to traditional distributed learning methods, FL is naturally more communication-efficient at scale (Konečný et al., 2016; Wang et al., 2019). Nonetheless, several issues stand out.

### 1.1 Motivation

*1) FL is susceptible to biased and low-quality local data.* Only a fraction of clients are selected for a round of FL (involving too many clients leads to diminishing gains (Li et al., 2019)). The standard FL algorithm (McMahan et al., 2017) selects clients randomly, which implies that every client (and its local data) is considered equally important. This makes the training process susceptible to local data with strong heterogeneity and of low quality (e.g., user-generated texts (Hard et al.,

---

[1]https://gdpr.eu/what-is-gdpr/

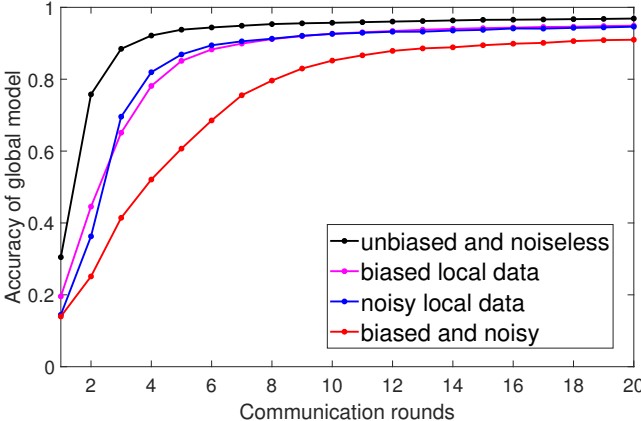

Figure 1: (Preliminary experiment) The global model's convergence under different data conditions. We ran the FL process with 100 clients to learn a CNN model on the MNIST dataset, which is partitioned and allocated to clients in four different ways where the data are 1) *original* (black line): noiseless and evenly distributed across the clients, 2) *biased* (magenta line): locally class-imbalanced, 3) *noisy* (blue line): blended with noise, or 4) *biased and noisy* (red line). The noise (if applied) covers 65% of the clients; the dominant class accounts for >50% of the samples for biased local data. The fraction of selected clients is 0.3 for each round.

2018) and noisy photos). In some scenarios, local data may contain irrelevant or even adversarial samples (Bhagoji et al., 2019; Bagdasaryan et al., 2020) from malicious clients (Fang et al., 2020; Bagdasaryan et al., 2020; Tolpegin et al., 2020). Traditional solutions such as data augmentation (Yoo et al., 2020) and re-sampling (Lin et al., 2017) prove useful for centralised training but applying them to local datasets may introduce extra noise (Cui et al., 2019) and increase the risk of information leakage (Yu et al., 2021). Another naive solution is to directly exclude those low-value clients with low-quality data, which, however, is often impractical because i) the quality of the data depends on the learning task and is difficult to gauge; ii) some noisy or biased data could be useful to the training at early stages (Feelders, 1996); and iii) sometimes low-quality data are very common across the clients.

In Fig. 1 we demonstrate the impact of involving "low-value" clients by running FL over 100 clients to learn a CNN model on MNIST using the standard FEDAVG algorithm. From the traces we can see that training over clients with problematic or strongly biased data can compromise the efficiency and efficacy of FL, resulting in an inferior global model that takes more rounds to converge.

*2) Learned representations can reflect data distribution and quality*. Representation learning is vital to the performance of deep models because learned representations can capture the intrinsic structure of data and provide useful information for the downstream machine learning tasks (Bengio et al., 2013). In ML research, The value of representations lies in the fact that they characterize the domain and learning task and provide task-specific knowledge (Morcos et al., 2018; Kornblith et al., 2019). In the context of FL, the similarity of representations are used for refining the model update rules (Li et al., 2021; Feng & Yu, 2020), but the distributional difference between representations of heterogeneous data is not yet explored.

Our study is also motivated by a key observation that representations from neural networks tend to have Gaussian patterns. As a demonstration we trained two different models (LeNet-5 and ResNet-18) on two different datasets (MNIST and CIFAR-100) separately. Fig. 2a shows the neuron-wise distribution of representations extracted from the first dense layer (FC-1) of LeNet-5. Fig. 2b shows the distribution of fused representations (in a channel-wise manner) extracted from a plain convolution layer and a residual block of ResNet-18.

These observations motivate us to study the distributional property of data representations and use it as a means to differentiate clients' value.

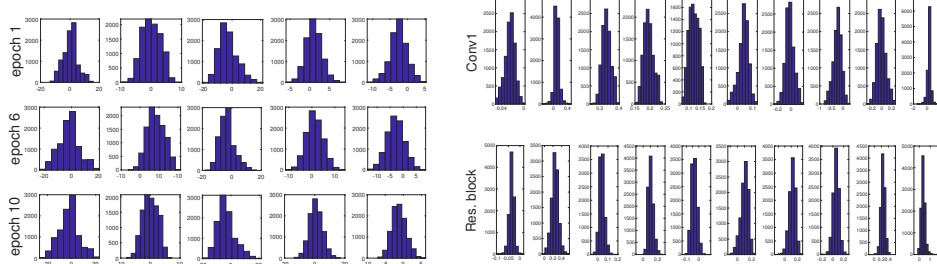

(a) Representations from FC-1 of a LeNet-5 model after being trained for 1, 6 and 10 epochs on MNIST.

(b) Fused representations from a standard convolution layer (1st row) and a residual block (2nd row) of a ResNet-18 model trained for 100 epochs on CIFAR-100.

Figure 2: (Preliminary experiment) Demonstration of learned representations from a distributional perspective. The representations are generated by forward propagation in model evaluation. Each box corresponds to a randomly sampled element in the representation vector.

## 1.2 CONTRIBUTIONS

Our contributions are summarized as the following:

- We first provide theoretical proof for the observation that data representations from neural networks tend to follow Gaussian distribution, based on which we propose a representation profiling and matching scheme for fast, low-cost comparison between different representation profiles.

- We present a novel FL algorithm FEDPROF that adaptively adjusts clients' participation probability based on representation profile dissimilarity.

- Results of extensive experiments show that FEDPROF reduces the number of communication rounds by up to 77%, shortens the overall training time (up to 4.5x speedup) while increasing the accuracy of the global model by up to 2.5%.

## 2 RELATED WORK

Different from traditional distributed training methods (e.g., Alistarh et al. (2017); Wu et al. (2018); Zheng et al. (2017)), Federated Learning assumes strict constraints of data locality and limited communication capacity (Konečnỳ et al., 2016). Much effort has been made in optimizing FL and covers a variety of perspectives including communication (Konečnỳ et al., 2016; Niknam et al., 2020; Cui et al., 2021), update rules (Li et al., 2020; Wu et al., 2021a; Leroy et al., 2019; Luping et al., 2019), flexible aggregation (Wang et al., 2019; Wu et al., 2021b) and personalization (Fallah et al., 2020; Tan et al., 2021; Deng et al., 2020).

The control of device participation is imperative in cross-device FL scenarios (Kairouz et al., 2019; Yang et al., 2020) where the quality of local data is uncontrollable and the clients show varied value for the training task (Tuor et al., 2020). To this end, the selection of clients is pivotal to the convergence of FL over heterogeneous data and devices (Nishio & Yonetani, 2019; Wang et al., 2020b; Chai et al., 2019; Acar et al., 2020). Non-uniform client selection is widely adopted in existing studies (Li et al., 2020; Goetz et al., 2019; Cho et al., 2020; Li et al., 2019; Chen et al., 2020b; Wang et al., 2020a) and has been theoretically proven with convergence guarantees (Chen et al., 2020b; Li et al., 2019). Many approaches sample clients based on their performance (Nishio & Yonetani, 2019; Chai et al., 2020) or aim to jointly optimize the model accuracy and training time (Shi et al., 2020; Chen et al., 2020a; 2021). A popular strategy is to use loss as the information to guide client selection (Goetz et al., 2019; Lai et al., 2021; Sarkar et al., 2020). For example, AFL (Goetz et al., 2019) prioritizes the clients with high loss feedback on local data, but it is potentially susceptible to noisy and unexpected data that yield illusive loss values. Data representations are

useful in the context of FL for information exchange (Feng & Yu, 2020) or objective adaptation. For example, Li et al. (2021) introduces representation similarities into local objectives. This contrastive learning approach guides local training to avoid model divergence. Nonetheless, the study on the distribution of data representations is still lacking whilst its connection to clients' training value is hardly explored either.

## 3 DATA REPRESENTATION PROFILING AND MATCHING

In this paper, we consider a typical cross-device FL setting (Kairouz et al., 2019), in which multiple end devices collaboratively perform local training on their own datasets $D_i, i = 1, 2, ..., n$. The server owns a validation dataset $D^*$ for model evaluation. Every dataset is only accessible to its owner.

Considering the distributional pattern of data representations (Fig. 2) and the role of the global model in FL, we propose to profile the representations of local data using the global model. In this section, we first provide theoretical proof to support our observation that representations from neural network models tend to follow Gaussian distribution. Then we present a novel scheme to profile data representations and define profile dissimilarity for fast and secure representation comparison.

### 3.1 GAUSSIAN DISTRIBUTION OF REPRESENTATIONS

We first make the following definition to facilitate our analysis.

**Definition 1** (The Lyapunov's condition). *A set of random variables $\{Z_1, Z_2, \ldots, Z_v\}$ satisfy the Lyapunov's condition if there exists a $\delta$ such that*

$$\lim_{v \to \infty} \frac{1}{s^{2+\delta}} \sum_{k=1}^{v} \mathrm{E}\left[|Z_k - \mu_k|^{2+\delta}\right] = 0, \tag{1}$$

*where $\mu_k = \mathrm{E}[Z_k]$, $\sigma_k^2 = \mathrm{E}[(Z_k - \mu_k)^2]$ and $s = \sqrt{\sum_{k=1}^{v} \sigma_k^2}$.*

The Lyapunov's condition can be intuitively explained as a limit on the overall variation (with $|Z_k - \mu_k|^{2+\delta}$ being the $(2 + \delta)$-th moment of $Z_k$) of a set of random variables.

Now we present Proposition 1 and Proposition 2. The Propositions provide theoretical support for our representation profiling and matching method to be introduced in Section 3.2.

**Proposition 1.** *The representations from linear operators (e.g., a pre-activation dense layer or a plain convolutional layer) in a neural network tend to follow the Gaussian distribution if the layer's weighted inputs satisfy the Lyapunov's condition.*

**Proposition 2.** *The fused representations[2] from non-linear operators (e.g., a hidden layer of LSTM or a residual block of ResNet) in a neural network tend to follow the Gaussian distribution if the layer's output elements satisfy the Lyapunov's condition.*

The proofs of Propositions 1 and 2 are provided in Appendices A.1 and A.2, respectively.

We base our proof on the Lyapunov's CLT which assumes independence between the variables. The assumption theoretically holds by using the Bayesian network concepts: let $X$ denote the layer's input and $H_k$ denote the $k$-th component in its output. The inference through the layer produces dependencies $X \to H_k$ for all $k$. According to Local Markov Property, we have $H_i$ independent of any $H_j$ ($j \neq i$) given $X$. Also, the Lyapunov's condition is typically met when the model is properly initialized and batch normalization is applied. Next, we discuss the proposed representation profiling and matching scheme.

### 3.2 DISTRIBUTIONAL PROFILING AND MATCHING

Based on the Gaussian pattern of representations, we compress the data representations statistically into a compact form called *representation profiles*. The profile produced by the global model $w$ on

---

[2]*Fused* representations refer to the sum of elements in the original representations produced by a single layer (channel-wise for a residual block).

a dataset $D$, denoted by $RP(w, D)$, has the following format:

$$RP(w, D) = \{\mathcal{N}(\mu_i, \sigma_i^2)\}_{i=1}^q, \tag{2}$$

where $q$ is the profile length determined by the dimensionality of the representations. For example, $q$ is equal to the number of kernels for channel-wise fused representations from a convolutional layer. The tuple $(\mu_i, \sigma_i^2)$ contains the mean and the variance of the $i$-th representation element.

Local representation profiles are generated by clients and sent to the server for comparison (the cost of transmission is negligible considering each profile is only $q \times 8$ bytes). Let $RP_k$ denote the local profile from client $k$, and $RP^*$ denote the baseline profile (generated in model evaluation) on the server. The dissimilarity between $RP_k$ and $RP^*$, denoted by $div(RP_k, RP^*)$, is defined as:

$$div(RP_k, RP^*) = \frac{1}{q} \sum_{i=1}^q \mathrm{KL}(\mathcal{N}_i^{(k)} || \mathcal{N}_i^*), \tag{3}$$

where $\mathrm{KL}(\cdot)$ denotes the Kullback–Leibler (KL) divergence. An advantage of our profiling scheme is that a much simplified KL divergence formula can be adopted because of the Gaussian distribution property (see Roberts & Penny (2002, Appendix B) for details), which yields:

$$\mathrm{KL}(\mathcal{N}_i^{(k)} || \mathcal{N}_i^*) = \log \frac{\sigma_i^*}{\sigma_i^{(k)}} + \frac{(\sigma_i^{(k)})^2 + (\mu_i^{(k)} - \mu_i^*)^2}{2(\sigma_i^*)^2}, \tag{4}$$

Eq. (4) computes the KL divergence without calculating any integral, which is computationally cost-efficient. Besides, the computation of profile dissimilarity can be performed under the Homomorphic Encryption for minimum knowledge disclosure (see Appendix D for details).

## 4 THE TRAINING ALGORITHM FEDPROF

Our research aims to optimize the global model over a large group of clients (datasets) of disparate training value. Given the client set $U(|U| = N)$, let $D_k$ denote the local dataset on client $k$ and $D^*$ the validation set on the server, We formulate the optimization problem in (5) where the coefficient $\rho_k$ differentiates the importance of the local objective functions $F_k(w)$ and depends on the data in $D_k$. Our global objective is in a sense similar to the agnostic learning scenario (Mohri et al., 2019) where a non-uniform mixture of local data distributions is implied.

$$\arg\min_w F(w) = \sum_{k=1}^N \rho_k F_k(w), \tag{5}$$

where $w$ is the parameter set of the global model. The coefficients $\{\rho_k\}_{k=1}^N$ add up to 1. $F_k(w)$ is client $k$'s local objective function of training based on the loss function $\ell(\cdot)$:

$$F_k(w) = \frac{1}{|D_k|} \sum_{(x_i, y_i) \in D_k} \ell(x_i, y_i; w), \tag{6}$$

Involving the "right" clients facilitates the convergence. With this motivation we score each client with $\lambda_k$ each round based on the representation profile dissimilarity:

$$\lambda_k = \exp\big(-\alpha \cdot div(RP_k(v_k), RP^*(v_k))\big), \tag{7}$$

where $v_k$ is the version of $RP_k$ (i.e., the version of the global model that client $k$ receives) and $\alpha$ is a preference factor deciding how biased the selection strategy needs to be towards the clients with small profile dissimilarity (i.e., higher training value). With $\alpha = 0$, our strategy is equivalent to random selection. The scores connect the representation profiling and matching scheme to the design of the selective client participation strategy adopted in our FL training algorithm FEDPROF, which is outlined in Algorithm 1 (see Appendix B for a detailed version with both client and server processes). The key steps of our algorithm are local representation profiling (line 5), baseline representation profiling (line 9) and client scoring (line 3). Fig. 3 illustrates the workflow of the proposed algorithm from representation profiling, matching to scheduling.

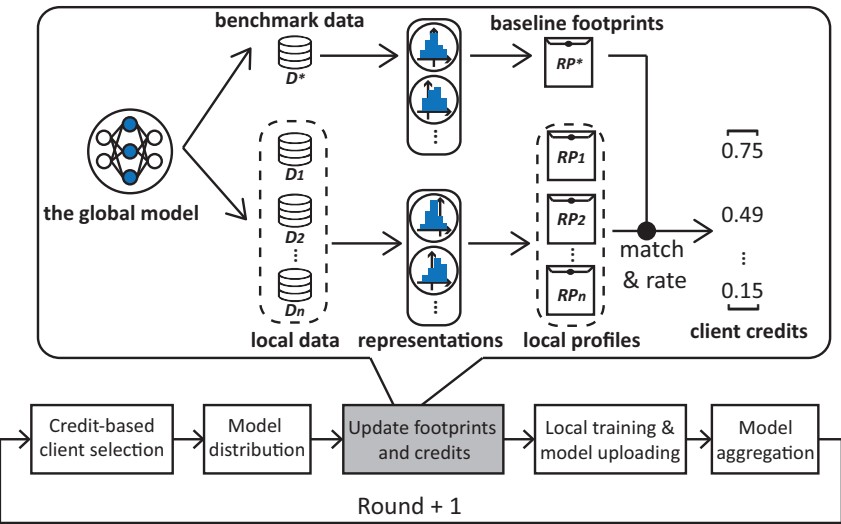

Figure 3: The workflow of the proposed FEDPROF algorithm.

---

**Algorithm 1:** the FEDPROF algorithm

**Input:** maximum number of rounds $T_{max}$, iterations per round $E$, fraction $C$

1 Initialize global model $w$ and generate baseline profile $RP^*$
2 Collect initial representation profiles $\{RP_k\}_{k \in U}$ from all clients

    **for** round $T \leftarrow 1$ to $T_{max}$ **do**

3     Update client scores $\{\lambda_k\}_{k \in U}$ and compute $\Lambda = \sum_{k \in U} \lambda_k$
4     $S \leftarrow$ Choose $K = N \cdot C$ clients by probability distribution $\{\frac{\lambda_k}{\Lambda}\}_{k \in U}$

        **for** client $k$ in $S$ **in parallel do**

5         $RP_k \leftarrow updateProfile(k, w, T - 1)$
6         $w_k \leftarrow localTraining(k, w, E)$

        **end**

7     Collect local profiles from the clients in $S$
8     Update $w$ via model aggregation
9     Evaluate $w$ and update $RP^*$

    **end**

10 return $w$

---

The convergence rate of FL algorithms with opportunistic client selection (sampling) has been extensively studied in the literature (Li et al., 2019; Chen et al., 2020b; Wang et al., 2019). Inspired by these studies, we present Theorem 1 to guarantee the global model's convergence for our algorithm. Similar to Stich et al. (2018); Zhang et al. (2013); Li et al. (2019), we assume that $\{F_k\}_{k=1}^N$ are $L$-smooth and $\mu$-strongly convex and that in expectation, the variance of local stochastic gradients are bounded by $\epsilon^2$ and their squared norms are bounded by $G^2$. For each local update step $t$ we define $S(t)$ as the set of selected clients in the associated round.

**Theorem 1.** *Using partial aggregation and a strategy $\pi$ that selects clients by the probability distribution $\{q_k\}_{k=1}^N$, the global model $w(t)$ converges in expectation by having $q_k = \rho_k$, an aggregation interval $E \geq 1$ and a decreasing step size (learning rate) $\eta_t = \frac{2}{\mu(t+\gamma)}$.*

$$\mathrm{E}\big[F(w(t))\big] - F^* \leq \frac{L}{(\gamma + t)} \Big( \frac{2(\mathcal{B} + \mathcal{C})}{\mu^2} + \frac{\gamma + 1}{2} \Delta_1 \Big), \tag{8}$$

*where $t \in T_A = \{nE | n = 1, 2, \ldots\}$, $\gamma = \max\{\frac{8L}{\mu}, E\} - 1$, $\mathcal{B} = \sum_{k=1}^N \rho_k^2 \epsilon_k^2 + 6L\Gamma + 8(E-1)^2 G^2$, $\mathcal{C} = \frac{4}{K} E^2 G^2$, $\Gamma = F^* - \sum_{k=1}^N \rho_k F_k^*$, $\Delta_1 = \mathrm{E}\|\bar{w}(1) - w^*\|^2$, $K = |S(t)| = N \cdot C$.*

The proof of Theorem 1 is provided in Appendix C.

## 5 EXPERIMENTS

We conducted extensive experiments to evaluate FEDPROF under various FL settings. Apart from FEDAVG (McMahan et al., 2017), we also reproduced several state-of-the-art FL algorithms for comparison. For fair comparison, the algorithms are grouped by the aggregation method (i.e., full aggregation and partial aggregation) and configured following the hyper-parameter settings in their papers (if any). Table 1 summarizes these FL algorithms. Note that our algorithm can adapt to both aggregation methods.

Table 1: The implemented FL algorithms for comparison

| Algorithm | Aggregation method | Rule of selection |
|---|---|---|
| FEDAVG (McMahan et al., 2017) | full aggregation | random selection |
| CFCFM (Wu et al., 2021b) | full aggregation | submission order |
| FEDAVG-RP (Li et al., 2019) | partial (Scheme II) | random selection |
| FEDPROX (Li et al., 2020) | partial aggregation | weighted random by data ratio |
| FEDADAM (Leroy et al., 2019) | partial with momentum | random selection |
| AFL (Goetz et al., 2019) | partial with momentum | local loss valuation |
| FEDPROF (ours) | full/partial aggregation | weighted random by score |

### 5.1 EXPERIMENT SETUP

We built our simulated FL system and implemented the algorithms based on the Pytorch framework (Build 1.7.0). We first set up a Small-scale Task (S-Task) to learn a multi-layer feed-forward network model from decentralized sensor data for predicting carbon monoxide (CO) and nitrogen oxides (NOx) emissions using the *GasTurbine*[3] dataset. Next, we set up a Large-scale Task (L-Task) to train a CNN over a large population of user devices for image classification (*EMNIST*[4]). In both tasks, data sharing is not allowed between any parties. The data are non-IID across the end devices.[5] We introduce a diversity of noise into the local datasets on end devices to simulate the discrepancy in data quality. The S-Task is performed over 50 sensor clients and 50% of them produce noisy data (including 10% invalid). The population for L-Task contains 1000 end devices across which the data spread with strong class imbalance – roughly 60% of the samples on each device fall into the same class. 15% of the local datasets in the L-Task are irrelevant images whereas another 45% of them are low-quality images (blurred or affected by salt-and-pepper noise). Considering the population of the clients, the setting of the selection fraction $C$ is based on the scale of the training participants suggested by Kairouz et al. (2019). For both tasks, the clients are heterogeneous in terms of both performance and communication bandwidth. More experimental settings are listed in Table 4 in Appendix E where the environment setup is also given in details.

### 5.2 EVALUATION RESULTS

We evaluate the performance of our FEDPROF algorithm in terms of the efficacy (best accuracy achieved) and efficiency (costs for convergence) in establishing a global model for the two tasks. Tables 2 and 3 report the average results of multiple runs with standard deviations for our algorithm. Figs. 4 and 5 plot the accuracy traces from the round-wise evaluations of the global model.

*1) Convergence in different aggregation modes:* Our results show a great difference in convergence rate under different aggregation modes. From Figs. 4 and 5, we observe that partial aggregation facilitates faster convergence of the global model than full aggregation, which is consistent with the observations made by Li et al. (2019). The advantage is especially obvious in the L-Task where partial aggregation requires much fewer communication rounds to reach the 90% accuracy. Our FEDPROF algorithm yields the fastest convergence in both groups of comparison for both tasks because selective participation benefits both aggregation methods.

---

[3]https://archive.ics.uci.edu/ml/datasets/Gas+Turbine+CO+and+NOx+Emission+Data+Set

[4]https://www.nist.gov/itl/products-and-services/emnist-dataset. We use the digits subset of EMNIST.

[5]In the S-Task, local datasets are of different sizes that follow a Gaussian distribution. In the L-Task, local data are largely imbalanced where around 60% of the samples on each client have the same class label.

Table 2: The results of running the S-Task. The best accuracy is achieved by running for long enough. Other metrics are recorded upon reaching the target accuracy (80% for the S-Task).

| | **Full aggregation** | | | | | | | |
| | **C=0.2** | | | | **C=0.3** | | | |
| | Best acc | For accuracy@0.8 | | | Best acc | For accuracy@0.8 | | |
| | | Rounds | Time(s) | E(Wh) | | Rounds | Time(s) | E(Wh) |
| FEDAVG | 0.805 | 56 | 2869.66 | 2.87 | 0.806 | 52 | 3160.01 | 4.12 |
| CFCFM | 0.806 | 39 | 1230.81 | 1.61 | 0.802 | 42 | 1495.34 | 2.91 |
| Ours | **0.824** | **16** | **803.74** | **0.80** | **0.827** | **12** | **701.18** | **0.94** |
| (std.) | 6.13E-03 | 2.06 | 139.76 | 0.14 | 8.01E-03 | 2.94 | 122.63 | 0.20 |
| | **Partial aggregation** | | | | | | | |
| | **C=0.2** | | | | **C=0.3** | | | |
| | Best acc | For accuracy@0.8 | | | Best acc | For accuracy@0.8 | | |
| | | Rounds | Time(s) | E(Wh) | | Rounds | Time(s) | E(Wh) |
| FEDAVG-RP | 0.819 | 13 | 735.48 | 0.71 | 0.817 | 8 | 466.90 | 0.64 |
| FEDPROX | 0.821 | 16 | 899.93 | 0.79 | 0.810 | 16 | 841.65 | 1.08 |
| FEDADAM | 0.818 | 8 | 438.20 | 0.42 | 0.819 | 12 | 667.00 | 0.94 |
| AFL | 0.816 | 6 | 313.81 | 0.30 | 0.813 | 6 | 298.81 | 0.42 |
| Ours | **0.844** | **5** | **283.76** | **0.27** | **0.841** | **4** | **235.22** | **0.35** |
| (std.) | 1.70E-03 | 0.47 | 30.69 | 0.03 | 7.07E-03 | 1.70 | 90.12 | 0.14 |

Table 3: The results of running the L-Task. The best accuracy is achieved by running for long enough. Other metrics are recorded upon reaching the target accuracy (90% for the L-Task).

| | **Full aggregation** | | | | | | | |
| | **C=0.05** | | | | **C=0.1** | | | |
| | Best acc | For accuracy@0.9 | | | Best acc | For accuracy@0.9 | | |
| | | Rounds | Time(s) | E(Wh) | | Rounds | Time(s) | E(Wh) |
| FEDAVG | 0.906 | 213 | 10407.30 | 60.01 | 0.929 | 76 | 3894.26 | 43.16 |
| CFCFM | 0.923 | 251 | 8645.17 | 61.37 | 0.932 | 75 | 2675.56 | 37.62 |
| Ours | **0.926** | **98** | **4846.15** | **28.22** | **0.945** | **45** | **2295.03** | **25.98** |
| (std.) | 8.16E-05 | 1.89 | 96.57 | 0.43 | 4.71E-04 | 0.82 | 29.46 | 0.01 |
| | **Partial aggregation** | | | | | | | |
| | **C=0.05** | | | | **C=0.1** | | | |
| | Best acc | For accuracy@0.9 | | | Best acc | For accuracy@0.9 | | |
| | | Rounds | Time(s) | E(Wh) | | Rounds | Time(s) | E(Wh) |
| FEDAVG-RP | 0.937 | 12 | 572.90 | 3.29 | 0.938 | 12 | 603.94 | 6.57 |
| FEDPROX | 0.936 | 13 | 640.01 | 3.58 | 0.942 | 11 | 559.16 | 5.91 |
| FEDADAM | 0.940 | 12 | 599.96 | 3.47 | 0.939 | 12 | 608.85 | 6.76 |
| AFL | 0.952 | 10 | 479.73 | 2.73 | 0.944 | 9 | 476.62 | 5.26 |
| Ours | **0.962** | **8** | **383.94** | **2.26** | **0.962** | **8** | **413.16** | **4.64** |
| (std.) | 9.43E-04 | 0.471 | 17.67 | 0.15 | 8.16E-04 | 0.47 | 11.64 | 0.03 |

*2) Best accuracy of the global model:* Through the training process of FL, the global model is evaluated each round on the server. The best global model obtained thus far is stored on the server. As shown in the 2nd column of Tables 2 and 3, our FEDPROF algorithm improves the accuracy achieved by 2.5% when compared against the baselines (FEDAVG and FEDAVG-RP). The AFL algorithm uses a loss-oriented client selection strategy, which shows the closest performance to our algorithm in the L-Task but the worst accuracy in the S-Task.

*3) Total communication rounds for convergence:* The number of communication rounds required for reaching convergence is a key indicator to the efficiency of FL. In the S-Task, our algorithm takes less than half the communication rounds required by other algorithms in most cases. In the L-Task with $C$=0.05, our algorithm reaches 90% accuracy within 100 rounds whilst FEDAVG and CFCFM need more than 200. In this case, FEDPROF also achieves approximately 7% higher accuracy in 50 rounds. Partial aggregation turns out to be much more communication-efficient: FEDAVG-RP

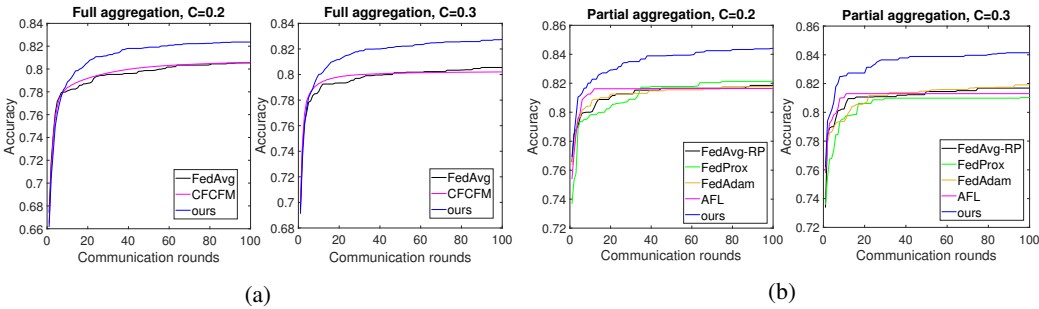

Figure 4: The traces of evaluation accuracy of the global model through 100 rounds in the S-Task using (a) the full-aggregation method, and (b) the partial-aggregation method.

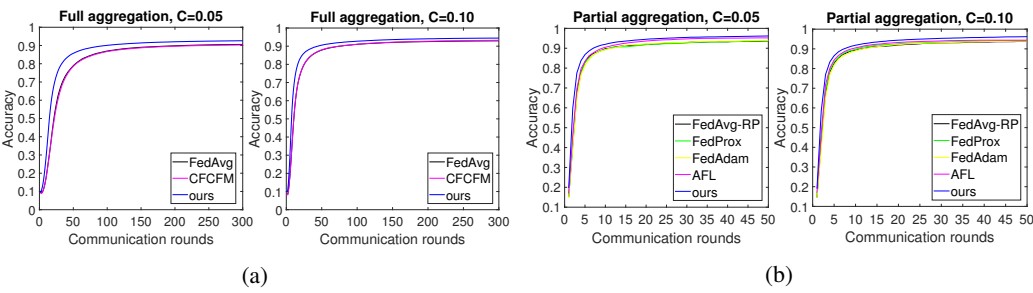

Figure 5: The traces of evaluation accuracy of the global model in the L-Task using (a) the full-aggregation method (through 300 rounds), and (b) the partial aggregation method (through 50 rounds).

needs 8 rounds to reach the accuracy target for the S-Task and 12 rounds for the L-Task, whilst our algorithm reduces the numbers to 5 and 8, respectively.

*4) Total time needed for convergence:* The overall time consumption is closely related to total communication rounds needed for convergence and the time cost for each round. Algorithms requiring more rounds to converge typically take longer to reach the accuracy target except the case of CFCFM, which priorities the clients that work faster. Using FEDAVG as the baseline, CFCFM accelerates the training process by 2.1x whilst our algorithm provides a 4.5x speedup for achieving the same target accuracy (S-Task, $C$=0.3). FEDPROF also has a clear advantage over FEDAVG-RP, FEDPROX and FEDADAM in the partial aggregation group where it shows a 2.6x speedup over FEDAVG-RP in the S-Task with $C$=0.2.

*5) Energy consumption of end devices:* A main concern for the end devices, as the participants of FL, is their power usage (Watt hours). Full aggregation methods experience slower convergence and thus endure higher energy cost on the devices. For example, with a small selection fraction $C$=0.05 in the L-Task, FEDAVG and CFCFM consume over 60Wh to reach the target accuracy. In this case, our algorithm reduces the cost by more than a half (28.22Wh). With the partial aggregation mode, the reduction by our algorithm is up to 62% (S-Task, $C = 0.2$). Considering all the cases, FEDPROF achieves the target accuracy with the least energy cost, providing an reduction of 29% $\sim$ 53%.

*6) Differentiated participation with* FEDPROF: Fig. 6 reflects the preference of our selection strategy. In the S-Task we can observe that the clients with useless samples or noisy data get significantly less involved ($<$10 on average) in training. In the L-Task our algorithm also effectively limits (basically excludes) the clients who owns the image data of poor quality (i.e., irrelevant or severely blurred), whereas the clients with moderately noisy images are selected with reduced frequency as compared to those with normal data. A potential issue of having the preference towards some of the devices is about fairness. Nonetheless, one can apply our algorithm together with an incentive mechanism (e.g., Yu et al. (2020)) to address it.

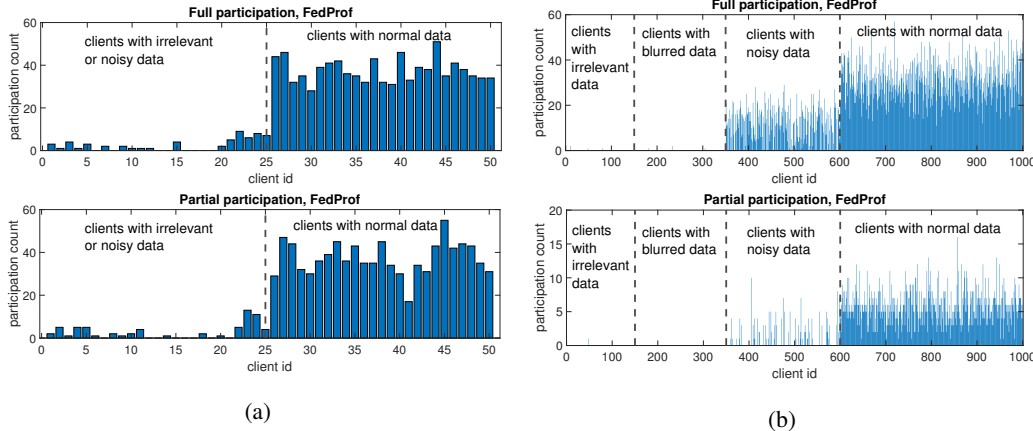

(a)                                                                (b)

Figure 6: Total counts by client of participation (i.e., being selected) in (a) the S-Task and (b) the L-Task. For clarity, clients are indexed according to their local data quality.

## 6 CONCLUSION

Federated learning provides a privacy-preserving approach to decentralized training but is vulnerable to the heterogeneity and uncertain quality of on-device data. In this paper, we use a novel approach to address the issue without violating the data locality restriction. We first provide key insights for the distribution of data representations and then develop a dynamic data representation profiling and matching scheme. Based on the scheme we propose a selective FL training algorithm FEDPROF that adaptively adjusts clients' participation chance based on their profile dissimilarity. We have conducted extensive experiments on public datasets under various environment settings. Evaluation results show that our algorithm significantly improves the efficiency of FL and reduces the time and energy costs for the global model to converge.

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

# A  PROOF OF PROPOSITIONS

## A.1  PROOF OF PROPOSITION 1

Without loss of generality, we provide the proof of Proposition 1 for the pre-activation representations from dense (fully-connected) layers and standard convolutional layers, respectively. The results can be easily extended to other linear neural operators.

*Dense layers*

*Proof.* Let $\Omega = \{neu_1, neu_2, ..., neu_q\}$ denote a dense layer (with $q$ neurons) of any neural network model and $H_k$ denote the pre-activation output of $neu_k$ in $\Omega$. We first provide the theoretical proof to support the observation that $H_k$ tends to follow the Gaussian distribution.

Let $\chi = \mathbb{R}^v$ denote the input feature space (with $v$ features) and assume the feature $X_i$ (which is a random variable) follows a certain distribution $\zeta_i(\mu_i, \sigma_i^2)$ (not necessarily Gaussian) with finite mean $\mu_i = \mathrm{E}[X_i]$ and variance $\sigma_i^2 = \mathrm{E}[X_i - \mu_i]$. For each neuron $neu_k$, let $\boldsymbol{w}_k = [w_{k,1} \, w_{k,2} \ldots w_{k,v}]$ denote the neuron's weight vector, $b_k$ denote the bias, and $Z_{k,i} = X_i w_{k,i}$ denote the $i$-th weighted input. Let $H_k$ denote the output of $neu_k$. During the forward propagation, we have:

$$
\begin{aligned}
H_k &= X \boldsymbol{w}_k^T + b_k \\
&= \sum_{i=1}^{v} X_i w_{k,i} + b_k \\
&= \sum_{i=1}^{v} Z_{k,i} + b_k.
\end{aligned}
\tag{9}
$$

Apparently $Z_{k,i}$ is a random variable because $Z_{k,i} = X_i w_{k,i}$ (where the weights $w_{k,i}$ are constants during a forward pass), thus $H_k$ is also a random variable according to Eq. (9).

In an ideal situation, the inputs variables $X_1, X_2, \ldots, X_v$ may follow a multivariate Gaussian distribution, in which case Proposition 1 automatically holds due to the property of multivariate normal distribution that every linear combination of the components of the random vector $(X_1, X_2, \ldots, X_v)^T$ follows a Gaussian distribution (Lemons, 2003). In other words, $H_k = X_1 w_{k,1} + X_2 w_{k,2} + \ldots + X_v w_{k,v} + b_k$ is a normally distributed variable since $w_{k,i}$ and $b_k$ ($k = 1, 2, \ldots, v$) are constants in the forward propagation. A special case for this condition is that $X_1, X_2, \ldots, X_v$ are independent on each other and $X_i$ follows a Gaussian distribution $\mathcal{N}(\mu_i, \sigma_i^2)$ for all $i = 1, 2, \ldots, v$. In this case, by the definition of $Z_{k,i}$, we have:

$$
Z_{k,i} = X_i w_{k,i} \sim \mathcal{N}\big(w_{k,i}\mu_i, (w_{k,i}\sigma_i)^2\big),
\tag{10}
$$

where $Z_1, Z_2, \ldots, Z_v$ are independent on each other. Combining Eqs. (9) and (10), we have:

$$
H_k \sim \mathcal{N}\Big( \sum_{i=1}^{v} w_{k,i}\mu_i + b_k, \sum_{i=1}^{v} (w_{k,i}\sigma_i)^2 \Big),
\tag{11}
$$

For more general cases where $X_1, X_2, \ldots, X_v$ are not necessarily normally distributed, we assume the weighted inputs $Z_{k,i}$ of the dense layer satisfy the Lyapunov's condition (see *definition* 1). As a result, we have the following according to the Central Limit Theorem (CLT) (Billingsley, 2008) considering that $X_i$ follows $\zeta_i(\mu_i, \sigma_i^2)$:

$$
\frac{1}{s_k} \sum_{i=1}^{v} \big( Z_{k,i} - w_{k,i}\mu_i \big) \xrightarrow{d} \mathcal{N}(0, 1)
\tag{12}
$$

where $s_k = \sqrt{\sum_{i=1}^{v} \big( w_{k,i}\sigma_i \big)^2}$ and $\mathcal{N}(0, 1)$ denotes the standard normal distribution. Equivalently, for every $neu_k$ we have:

$$
\sum_{i=1}^{v} Z_{k,i} \xrightarrow{d} \mathcal{N}\big( \sum_{i=1}^{v} w_{k,i}\mu_i, s_k^2 \big)
\tag{13}
$$

Combining Eqs. (9) and (13) we can derive that:

$$H_k \xrightarrow{d} \mathcal{N}(\sum_{i=1}^{v} w_{k,i}\mu_i + b_k, s_k^2), \tag{14}$$

which means that $H_k(k = 1, 2, \ldots, v)$ tend to follow Gaussian distribution and proves our Proposition 1 for fully-connected layers. $\square$

*Convolutional layers*

*Proof.* Standard convolution in CNNs is also a linear transformation of the input feature space and its main difference from dense layers rests on the restricted size of receptive field. Without loss of generality, we analyze the representation (output) of a single kernel. To facilitate our analysis for convolutional layers, let $C$ denote the number of input channels and $K$ denote the kernel size. For ease of presentation, we define a receptive field mapping function $\Theta(k, i, j)$ that maps the positions ($k$ for channel index, $i$ and $j$ for indices on the same channel) of elements in the feature map (i.e., the representations) to the input features. For the $k$-th kernel, let $W_k$ denote its weight tensor (with $W_{k,c}$ being the weight matrix for channel $c$) and $b_k$ its bias.

Given the corresponding input patch $X_{\Theta(k,i,j)}$, The element $H_{k,i,j}$ of the representations from a convolutional layer can be formulated as:

$$H_{k,i,j} = \sum_{c=1}^{C} \sum_{i'=1}^{K} \sum_{j'=1}^{K} \left( X_{\Theta(k,i,j)} \circ W_{k,c} \right)_{i',j'} + b_k, \tag{15}$$

where $\circ$ denotes Hadamard product. The three summations reduce the results of element-wise product between the input patch and the $k$-th kernel to the correspond representation element $H_{k,i,j}$ in the feature map. For ease of presentation, here we use the notation $Z_{c,i',j'}^{(k)}$ to replace $\left( X_{\Theta(k,i,j)} \circ W_{k,c} \right)_{i',j'}$ and let $\zeta(\mu_{c,i',j'}, \sigma_{c,i',j'}^2)$ be the distribution that $Z_{c,i',j'}^{(k)}$ follows. Note that $\zeta$ can be any distribution since we do not make any distributional assumption on $Z_{c,i',j'}^{(k)}$.

With the notations, Eq. (15) can be rewritten in a similar form to Eq. (9):

$$H_{k,i,j} = \sum_{c=1}^{C} \sum_{i'=1}^{K} \sum_{j'=1}^{K} Z_{c,i',j'}^{(k)} + b_k. \tag{16}$$

We use the condition that the random variables $Z_{c,i',j'}^{(k)}$ satisfy the Lyapunov's condition, i.e., there exists a $\delta$ such that

$$\lim_{C \times K^2 \to \infty} \frac{1}{s^{2+\delta}} \sum_{c=1}^{C} \sum_{i'=1}^{K} \sum_{j'=1}^{K} \mathrm{E}\left[ |Z_{c,i',j'}^{(k)} - \mu_{c,i',j'}|^{2+\delta} \right] = 0, \tag{17}$$

where $s = \sqrt{\sum_{c=1}^{C} \sum_{i'=1}^{K} \sum_{j'=1}^{K} \sigma_{c,i',j'}^2}$.

Then according to the Lyapunov CLT, the following holds:

$$H_{k,i,j} \xrightarrow{d} \mathcal{N}(\sum_{c,i',j' \in \Theta(k,i,j)} \mu_{c,i',j'} + b_k, \sum_{c,i',j' \in \Theta(k,i,j)} \sigma_{c,i',j'}^2), \tag{18}$$

which proves our Proposition 1 for standard convolution layers. $\square$

### A.2 PROOF OF PROPOSITION 2

Without loss of generality, we prove Proposition 2 for the fused representations from the LSTM layer and the residual block of ResNet models, respectively. The results can be easily extended to other non-linear neural operators.

*LSTM*

*Proof.* Long Short-Term Memory (LSTM) models are popular for extracting useful representations from sequence data for tasks such as speech recognition and language modeling. Each LSTM layer contains multiple neural units. For the $k$-th unit, it takes as input the current input feature vector $X_t = (X_{t,1}, X_{t,2}, \ldots)$, hidden state vector $H_{t-1}$ and its cell state $c_{t-1,k}$. The outputs of the unit are its new hidden state $h_{t,k}$ and cell state $c_{t,k}$. In this paper, we study the distribution of $h_{t,k}$. Multiple gates are adopted in an LSTM unit: by $i_{t,k}$, $f_{t,k}$, $g_{t,k}$ and $o_{t,k}$ we denote the input gate, forget gate, cell gate and output gate of the LSTM unit $k$ at time step $t$. The update rules of these gates and the cell state are:

$$
\begin{aligned}
i_{t,k} &= \mathrm{sigmoid}(W_{(i)k}[H_{t-1}, X_t] + b_{(i)k}), \\
f_{t,k} &= \mathrm{sigmoid}(W_{(f)k}[H_{t-1}, X_t] + b_{(f)k}), \\
g_{t,k} &= \tanh(W_{(g)k}[H_{t-1}, X_t] + b_{(g)k}), \\
o_{t,k} &= \mathrm{sigmoid}(W_{(o)k}[H_{t-1}, X_t] + b_{(o)k}), \\
c_{t,k} &= f_{t,k} \cdot c_{t-1,k} + i_{t,k} \cdot g_{t,k},
\end{aligned}
\tag{19}
$$

where the $W_{(i)k}$, $W_{(f)k}$, $W_{(g)k}$ and $W_{(o)k}$ are the weight parameters and $b_{(i)k}$, $b_{(f)k}$, $b_{(g)k}$ and $b_{(o)k}$ are the bias parameters for the gates.

The output of the LSTM unit $h_{t,k}$ is calculated as the following:

$$
h_{t,k} = o_{t,k} \cdot \tanh(c_{t,k}).
\tag{20}
$$

Using the final hidden states $h_{T,k}$ (with $T$ being the length of the sequence) as the elements of the layer-wise representation, we apply the following layer-wise fusion to further produce $H$ over all the $h_{T,k}$ in a single LSTM layer:

$$
H = \sum_{k=1}^{d} h_{T,k},
\tag{21}
$$

where $d$ is the dimension of the LSTM layer. Again, by $\zeta(\mu_k, \sigma_k^2)$ we denote the distribution of $h_{T,k}$ (where the notation $T$ is dropped here since it is typically a fixed parameter). With $\{h_{T,k} | k = 1, 2, \ldots, d\}$ satisfying the Lyapunov's condition and by Central Limit Theorem $H$, tends to follow the Gaussian distribution:

$$
H \xrightarrow{d} \mathcal{N}(\sum_{i=1}^{d} \mu_k, \sum_{i=1}^{d} \sigma_k^2),
\tag{22}
$$

which proves the Proposition 2 for layer-wise fused representations from LSTM. $\square$

*Residual blocks*

*Proof.* Residual blocks are the basic units in the Residual neural network (ResNet) architecture (He et al., 2016). A typical residual block contains two convolutional layers with batch normalization (BN) and uses the ReLU activation function. The input of the whole block is added to the output of the second convolution (after BN) through a skip connection before the final activation. Since the convolution operators are the same as we formulate in the second part of Section A.1, here we use the notation $\Psi(X)$ to denote the sequential operations of convolution on $X$ followed by BN, i.e., $\Psi(X) \triangleq BN(Conv(X))$. Again, we reuse the receptive field mapping $\Theta(k, i, j)$ as defined in Section A.1 to position the inputs of the residual block corresponding to the element $Z_{k,i,j}$ in the output representation of the whole residual block.

Let $X$ denote the input of the residual block and $Z_{k,i,j}$ denote an element in the output tensor of the whole residual block. Then we have:

$$
\begin{aligned}
Z_{k,i,j} &= f\Big(X_{k,i,j} + BN\big(Conv\big(f\big(BN(Conv(X_{\Theta(k,i,j)}))\big)\big)\big)\Big) \\
&= f\Big(X_{k,i,j} + \Psi\big(f(\Psi(X_{\Theta(k,i,j)}))\big)\Big),
\end{aligned}
\tag{23}
$$

where $f$ is the activation function (ReLU).

We perform channel-wise fusion on the representation from the residual block to produce $H_k$ for the $k$-th channel:

$$H_k = \sum_{i=1}^{d_H} \sum_{j=1}^{d_W} Z_{k,i,j}, \tag{24}$$

where $d_H$ and $d_W$ are the dimensions of the feature map and $k$ is the channel index.

Let $\zeta(\mu_{k,i,j}, \sigma_{k,i,j}^2)$ denote the distribution that $Z_{k,i,j}$ follows. Then we apply the Lyapunov's condition to the representation elements layer-wise, i.e.,

$$\lim_{d_W \times d_H \to \infty} \frac{1}{s_k^{2+\delta}} \sum_{i=1}^{d_H} \sum_{j=1}^{d_W} \mathrm{E}\left[ |Z_{k,i,j} - \mu_{k,i,j}|^{2+\delta} \right] = 0, \tag{25}$$

where $s_k = \sqrt{\sum_{i=1}^{d_H} \sum_{j=1}^{d_W} \sigma_{k,i,j}^2}$.

With the above condition satisfied, by CLT $H_k$ (the fused representation on channel $k$) tends to follow the Gaussian distribution:

$$H_k \xrightarrow{d} \mathcal{N}(\sum_{i=1}^{d_H} \sum_{j=1}^{d_W} \mu_{k,i,j}, \sum_{i=1}^{d_H} \sum_{j=1}^{d_W} \sigma_{k,i,j}^2), \tag{26}$$

which proves the Proposition 2 for channel-wise fused representations from any residual block. □

## B  PSEUDO-CODE OF FEDPROF (DETAILED VERSION)

See Algorithm 2 for a complete version of the proposed training algorithm.

## C  CONVERGENCE ANALYSIS

In this section we provide the proof of the proposed Theorem 1. The analysis is mainly based on the results provided by Li et al. (2019). We first introduce several notations to facilitate the analysis.

### C.1  NOTATIONS

Let $U$ ($|U| = N$) denote the full set of clients and $S(t)$ ($|S(t)| = K$) denote the set of clients selected for participating. By $w_k(t)$ we denote the local model on client $k$ at time step $t$. We define an auxiliary sequence $v_k(t)$ for each client to represent the immediate local model after a local SGD update. Note that $v_k(t)$ is updated from $v_k(t-1)$ with learning rate $\eta_{t-1}$:

$$v_k(t) = w_k(t-1) - \eta_{t-1} \nabla F_k(w_k(t-1), \xi_{k,t-1}), \tag{27}$$

where $\nabla F_k(w_k(t-1), \xi_{k,t-1})$ is the stochastic gradient computed over a batch of data $\xi_{k,t-1}$ drawn from $D_k$ with regard to $w_k(t-1)$.

We also define two virtual sequences $\bar{v}(t) = \sum_{k=1}^{N} \rho_k v_k(t)$ and $\bar{w}(t) = Aggregate(\{v_k(t)\}_{k \in S(t)})$ for every time step $t$ (Note that the actual global model $w(t)$ is only updated at the aggregation steps $T_A = \{E, 2E, 3E, \ldots\}$). Given an aggregation interval $E \geq 1$, we provide the analysis for the partial aggregation rule that yields $\bar{w}(t)$ as:

$$\bar{w}(t) = \frac{1}{K} \sum_{k \in S(t)} v_k(t), \tag{28}$$

where $S(t)$ ($|S(t)| = K$) is the selected set of clients for the round $\lceil \frac{t}{E} \rceil$ that contains step $t$. At the aggregation steps $T_A$, $w(t)$ is equal to $\bar{w}(t)$, i.e., $w(t) = \bar{w}(t)$ if $t \in T_A$.

To facilitate the analysis, we assume each client always performs model update (and synchronization) to produce $v_k(t)$ and $\bar{v}(t)$ (but obviously it does not affect the resulting $\bar{w}$ and $w$ for $k \notin S(t)$).

$$w_k(t) = \begin{cases} v_k(t), & \text{if } t \notin T_A \\ \bar{w}(t), & \text{if } t \in T_A \end{cases} \tag{29}$$

---

**Algorithm 2:** the FEDPROF protocol (detailed version)

---

**Input** : maximum number of rounds $T_{max}$, local iterations per round $E$, client set $U$, client
fraction $C$, validation set $D^*$;

**Output:** the global model $w$

**// Server process: running on the server**

1 Initialize global model $w$ using a seed
2 $v \leftarrow 0$ // version of the latest global model
3 Broadcast the seed to all clients for identical model initialization
4 Collect initial profiles $\{RP_k\}_{k \in U}$ from all the clients
5 $v_k \leftarrow 0, \forall k \in U$
6 Generate initial baseline profile $RP^*(0)$ on $D^*$
7 $K \leftarrow |U| \cdot C$
**for** round $T \leftarrow 1$ to $T_{max}$ **do**
8     Calculate $div(RP_k(v_k), RP^*(v_k))$ for each client $k$
9     Update client scores $\{\lambda_k\}_{k \in U}$ and compute $\Lambda = \sum_{k \in U} \lambda_k$
10     $S \leftarrow$ Choose $K$ clients by probability distribution $\{\frac{\lambda_k}{\Lambda}\}_{k \in U}$
11     Distribute $w$ to the clients in $S$
    **for** client $k$ in $S$ **in parallel do**
12        $v_k \leftarrow v, \forall k \in S$
13        $RP_k(v_k) \leftarrow updateProfile(k, w, v)$
14        $w_k \leftarrow localTraining(k, w, E)$
    **end**
15     Collect local profiles from the clients in $S$
16     Update $w$ via model aggregation
17     $v \leftarrow T$
18     Evaluate $w$ and generate $RP^*(v)$
**end**
19 return $w$

**// Client process: running on client $k$**
**updateProfile($k, w, v$):**
20     Generate $RP_k$ on $D_k$ with the global $w$ received
21     Label profile $RP_k$ with version number $v$
22     Return $RP_k$
**return**
**localTraining($k, w, E$):**
23     $w_k \leftarrow w$
    **for** step $e \leftarrow 1$ to $E$ **do**
24        Update $w_k$ using gradient-based method
    **end**
25     Return $w_k$
**return**

---

For ease of presentation, we also define two virtual gradient sequences: $\bar{g}(t) = \sum_{k=1}^{N} \rho_k \nabla F_k(w_k(t))$ and $g(t) = \sum_{k=1}^{N} \rho_k \nabla F_k(w_k(t), \xi_{k,t})$. Thus we have $\mathrm{E}[g(t)] = \bar{g}(t)$ and $\bar{v}(t) = \bar{w}(t-1) - \eta_{t-1}g(t-1)$.

## C.2 ASSUMPTIONS

We formally make four assumptions to support our analysis of convergence. Assumptions 1 and 2 are standard in the literature (Li et al., 2019; Wang et al., 2019; Cho et al., 2020) defining the convexity and smoothness properties of the objective functions. Assumptions 3 and 4 bound the variance of the local stochastic gradients and their squared norms in expectation, respectively. These two assumptions are also made in by Li et al. (2019).

**Assumption 1.** *$F_1, F_2, \ldots, F_N$ are L-smooth, i.e., for any $k \in U$, x and y: $F_k(y) \leq F_k(x) + (y - x)^T \nabla F_k(x) + \frac{L}{2}\|y - x\|_2^2$*

It is obvious that the global objective $F$ is also $L$-smooth as a linear combination of $F_1, F_2, \ldots, F_N$ with $\rho_1, \rho_2, \ldots, \rho_N$ being the weights.

**Assumption 2.** *$F_1, F_2, \ldots, F_N$ are $\mu$-strongly convex, i.e., for all $k \in U$ and any $x$, $y$: $F_k(y) \geq F_k(x) + (y-x)^T \nabla F_k(x) + \frac{\mu}{2}\|y-x\|_2^2$*

**Assumption 3.** *The variance of local stochastic gradients on each device is bounded: For all $k \in U$, $\mathrm{E}\|\nabla F_k(w_k(t), \xi_{t,k}) - F_k(w_k(t))\|^2 \leq \epsilon^2$*

**Assumption 4.** *The squared norm of local stochastic gradients on each device is bounded: For all $k \in U$, $\mathrm{E}\|\nabla F_k(w_k(t), \xi_{t,k})\|^2 \leq G^2$*

## C.3 KEY LEMMAS

To facilitate the proof of our main theorem, we first present several key lemmas.

**Lemma 1** (Result of one SGD step). *Under Assumptions 1 and 2 and with $\eta_t < \frac{1}{4L}$, for any $t$ it holds true that*

$$\mathrm{E}\|\bar{v}(t+1) - w^*\|^2 \leq (1-\eta_t\mu)\mathrm{E}\|\bar{w}(t) - w^*\|^2 + \eta_t^2\mathrm{E}\|g_t - \bar{g}_t\|^2 + 6L\eta_t^2\Gamma + 2\mathrm{E}\Big[\sum_{k=1}^N \rho_k\|w_k(t) - \bar{w}(t)\|^2\Big],$$

(30)

*where $\Gamma = F^* - \sum_{k=1}^N \rho_k F_k^*$.*

**Lemma 2** (Gradient variance bound). *Under Assumption 3, one can derive that*

$$\mathrm{E}\|g_t - \bar{g}_t\|^2 \leq \sum_{k=1}^N \rho_k^2 \epsilon_k^2.$$

(31)

**Lemma 3** (Bounded divergence of $w_k(t)$). *Assume Assumption 4 holds and a non-increasing step size $\eta_t$ s.t. $\eta_t \leq 2\eta_{t+E}$ for all $t = 1, 2, \ldots$, it follows that*

$$\mathrm{E}\Big[\sum_{k=1}^N \rho_k\|w_k(t) - \bar{w}(t)\|^2\Big] \leq 4\eta_t^2(E-1)^2 G^2.$$

(32)

Lemmas 1, 2 and 3 hold for both full and partial participation and are independent of the client selection strategy. We refer the readers to Li et al. (2019) for their proofs and focus our analysis on opportunistic selection.

The next two lemmas give important properties of the aggregated model $\bar{w}$ as a result of partial participation and non-uniform client selection/sampling.

**Lemma 4** (Unbiased aggregation). *For any aggregation step $t \in T_A$ and with $q_k = \rho_k$ in the selection of $S(t)$, it follows that*

$$\mathrm{E}_{S(t)}[\bar{w}(t)] = \bar{v}(t).$$

(33)

*Proof.* First, we present a key observation given by Li et al. (2019) as an important trick to handle the randomness caused by client selection with probability distribution $\{q_k\}_{k=1}^N$. By taking the expectation over $S(t)$, it follows that

$$\mathrm{E}_{S(t)} \sum_{k \in S(t)} X_k = K \mathrm{E}_{S(t)}[X_k] = K \sum_{k=1}^N q_k X_k.$$

(34)

Let $q_k = \rho_k$, take the expectation of $\bar{w}(t)$ over $S(t)$ and notice that $\bar{v}(t) = \sum_{k \in U} \rho_k v_k(t)$:

$$
\begin{aligned}
\mathrm{E}_{S(t)}[\bar{w}(t)] &= \mathrm{E}_{S(t)}\Big[\frac{1}{K} \sum_{k \in S(t)} v_k(t)\Big] \\
&= \frac{1}{K} \mathrm{E}_{S(t)}\Big[ \sum_{k \in S(t)} [v_k(t)\Big] \\
&= \frac{1}{K} K \mathrm{E}_{S(t)}[v_k(t)] \\
&= \sum_{k \in U} q_k v_k(t) \\
&= \bar{v}(t).
\end{aligned}
$$

$\square$

**Lemma 5** (Bounded variance of $\bar{w}(t)$). *For any aggregation step $t \in T_A$ and with a non-increasing step size $\eta_t$ s.t. $\eta_t \leq 2\eta_{t+E-1}$, it follows that*

$$
\mathrm{E}_{S(t)}\|\bar{w}(t) - \bar{v}(t)\|^2 \leq \frac{4}{K}\eta_{t-1}^2 E^2 G^2. \tag{35}
$$

*Proof.* First, one can prove that $v_k(t)$ is an unbiased estimate of $\bar{v}(t)$ for any $k$:

$$
\mathrm{E}_{S(t)}[v_k(t)] = \sum_{k \in U} q_k v_k(t) = \bar{v}(t). \tag{36}
$$

Then by the aggregation rule $\bar{w}(t) = \frac{1}{K} \sum_{k \in S(t)} v_k(t)$, we have:

$$
\begin{aligned}
\mathrm{E}_{S(t)}\|\bar{w}(t) - \bar{v}(t)\|^2 &= \frac{1}{K^2}\mathrm{E}_{S(t)}\|K\bar{w}(t) - K\bar{v}(t)\|^2 \\
&= \frac{1}{K^2}\mathrm{E}_{S(t)}\Big\| \sum_{k \in S(t)} v_k(t) - \sum_{k=1}^{K} \bar{v}(t)\Big\|^2 \\
&= \frac{1}{K^2}\mathrm{E}_{S(t)}\Big\| \sum_{k \in S(t)} \big(v_k(t) - \bar{v}(t)\big)\Big\|^2 \\
&= \frac{1}{K^2}\Big(\mathrm{E}_{S(t)} \sum_{k \in S(t)} \|v_k(t) - \bar{v}(t)\|^2 \\
&\quad + \mathrm{E}_{S(t)} \underbrace{\sum_{i,j \in S(t), i \neq j} \langle v_i(t) - \bar{v}(t), v_j(t) - \bar{v}(t)\rangle}_{=0}\Big), \tag{37}
\end{aligned}
$$

where the second term on the RHS of (37) equals zero because $\{v_k(t)\}_{k \in U}$ are independent and unbiased (see Eq. 36). Further, by noticing $t - E \in T_A$ (because $t \in T_A$) which implies that $w_k(t - E) = \bar{w}(t - E)$ since the last communication, we have:

$$
\begin{aligned}
\mathrm{E}_{S(t)}\|\bar{w}(t) - \bar{v}(t)\|^2 &= \frac{1}{K^2}\mathrm{E}_{S(t)} \sum_{k \in S(t)} \|v_k(t) - \bar{v}(t)\|^2 \\
&= \frac{1}{K^2}K\mathrm{E}_{S(t)}\|v_k(t) - \bar{v}(t)\|^2 \\
&= \frac{1}{K}\mathrm{E}_{S(t)}\|\big(v_k(t) - \bar{w}(t - E)\big) - \big(\bar{v}(t) - \bar{w}(t - E)\big)\|^2 \\
&\leq \frac{1}{K}\mathrm{E}_{S(t)}\|v_k(t) - \bar{w}(t - E)\|^2, \tag{38}
\end{aligned}
$$

where the last inequality results from $\mathrm{E}[v_k(t) - \bar{w}(t-E)] = \bar{v}(t) - \bar{w}(t-E)$ and that $\mathrm{E}\|X - \mathrm{E}X\|^2 \leq \mathrm{E}\|X\|^2$. Further, we have:

$$
\begin{aligned}
\mathrm{E}_{S(t)}\|\bar{w}(t) - \bar{v}(t)\|^2 &\leq \frac{1}{K}\mathrm{E}_{S(t)}\|v_k(t) - \bar{w}(t-E)\|^2 \\
&= \frac{1}{K}\sum_{k=1}^{N} q_k \mathrm{E}_{S(t)}\|v_k(t) - \bar{w}(t-E)\|^2 \\
&= \frac{1}{K}\sum_{k=1}^{N} q_k \underbrace{\mathrm{E}_{S(t)}\|\sum_{i=t-E}^{t-1}\eta_i\nabla F_k(w_k(i),\xi_{k,i})\|^2}_{Z_1}.
\end{aligned}
\tag{39}
$$

Let $i_m = \arg\max_i \|\nabla F_k(w_k(i),\xi_{k,i})\|, i \in [t-E, t-1]$. By using the Cauchy-Schwarz inequality, Assumption 4 and choosing a non-increasing $\eta_t$ s.t. $\eta_t \leq 2\eta_{t+E-1}$, we have:

$$
\begin{aligned}
Z_1 &= \mathrm{E}_{S(t)}\|\sum_{i=t-E}^{t-1}\eta_i\nabla F_k(w_k(i),\xi_{k,i})\|^2 \\
&= \sum_{i=t-E}^{t-1}\sum_{j=t-E}^{t-1}\mathrm{E}_{S(t)}\langle\eta_i\nabla F_k(w_k(i),\xi_{k,i}), \eta_j\nabla F_k(w_k(j),\xi_{k,j})\rangle \\
&\leq \sum_{i=t-E}^{t-1}\sum_{j=t-E}^{t-1}\mathrm{E}_{S(t)}\Big[\|\eta_i\nabla F_k(w_k(i),\xi_{k,i})\| \cdot \|\eta_j\nabla F_k(w_k(j),\xi_{k,j})\|\Big] \\
&\leq \sum_{i=t-E}^{t-1}\sum_{j=t-E}^{t-1}\eta_i\eta_j \cdot \mathrm{E}_{S(t)}\|\nabla F_k(w_k(i_m),\xi_{k,i_m})\|^2 \\
&\leq \sum_{i=t-E}^{t-1}\sum_{j=t-E}^{t-1}\eta_{t-E}^2 \cdot \mathrm{E}_{S(t)}\|\nabla F_k(w_k(i_m),\xi_{k,i_m})\|^2 \\
&\leq 4\eta_{t-1}^2 E^2 G^2.
\end{aligned}
\tag{40}
$$

Plug $Z_1$ back into (39) and notice that $\sum_{k=1}^{N} q_k = 1$, we have:

$$
\begin{aligned}
\mathrm{E}_{S(t)}\|\bar{w}(t) - \bar{v}(t)\|^2 &\leq \frac{1}{K}\sum_{k=1}^{N} q_k 4\eta_t^2 E^2 G^2 \\
&= \frac{4}{K}\eta_{t-1}^2 E^2 G^2.
\end{aligned}
$$

$\square$

### C.4 PROOF OF THEOREM 1

*Proof.* Taking expectation of $\|\bar{w}(t) - w^*\|^2$, we have:

$$
\begin{aligned}
\mathrm{E}\|\bar{w}(t) - w^*\|^2 &= \mathrm{E}_{S(t)}\|\bar{w}(t) - \bar{v}(t) + \bar{v}(t) - w^*\|^2 \\
&= \underbrace{\mathrm{E}\|\bar{w}(t) - \bar{v}(t)\|^2}_{A_1} + \underbrace{\mathrm{E}\|\bar{v}(t) - w^*\|^2}_{A_2} + \underbrace{\mathrm{E}\langle\bar{w}(t) - \bar{v}(t), \bar{v}(t) - w^*\rangle}_{A_3}
\end{aligned}
\tag{41}
$$

where $A_3$ vanishes because $\bar{w}(t)$ is an unbiased estimate of $\bar{v}(t)$ by first taking expectation over $S(t)$ (Lemma 4).

To bound $A_2$ for $t \in T_A$, we apply Lemma 1:

$$A_2 = \mathrm{E}\|\bar{v}(t) - w^*\|^2 \le (1 - \eta_{t-1}\mu)\mathrm{E}\|\bar{w}(t-1) - w^*\|^2 + \underbrace{\eta_{t-1}^2\mathrm{E}\|g_{t-1} - \bar{g}_{t-1}\|^2}_{B_1}$$

$$+ 6L\eta_{t-1}^2\Gamma + \underbrace{\mathrm{E}\Big[\sum_{k=1}^{N} \rho_k\|w_k(t-1) - \bar{w}(t-1)\|^2\Big]}_{B_2}. \tag{42}$$

Then we use Lemmas 2 and 3 to bound $B_1$ and $B_2$ respectively, which yields:

$$A_2 = \mathrm{E}\|\bar{v}(t) - w^*\|^2 \le (1 - \eta_{t-1}\mu)\mathrm{E}\|\bar{w}(t-1) - w^*\|^2 + \eta_{t-1}^2\mathcal{B}, \tag{43}$$

where $\mathcal{B} = \sum_{k=1}^{N} \rho_k^2\epsilon_k^2 + 6L\Gamma + 8(E-1)^2G^2$.

To bound $A_1$, one can first take expectation over $S(t)$ and apply Lemma 5 where the upper bound actually eliminates both sources of randomness. Thus, it follows that

$$A_1 = \mathrm{E}\|\bar{w}(t) - \bar{v}(t)\|^2 \le \frac{4}{K}\eta_{t-1}^2E^2G^2 \tag{44}$$

Let $\mathcal{C} = \frac{4}{K}E^2G^2$ and plug $A_1$ and $A_2$ back into (41):

$$\mathrm{E}\|\bar{w}(t) - w^*\|^2 \le (1 - \eta_{t-1}\mu)\mathrm{E}\|\bar{w}(t-1) - w^*\|^2 + \eta_{t-1}^2(\mathcal{B} + \mathcal{C}). \tag{45}$$

Equivalently, let $\Delta_t = \mathrm{E}\|\bar{w}(t) - w^*\|^2$, then we have the following recurrence relation for any $t \ge 1$:

$$\Delta_t \le (1 - \eta_{t-1}\mu)\Delta_{t-1} + \eta_{t-1}^2(\mathcal{B} + \mathcal{C}). \tag{46}$$

Next we prove by induction that $\Delta_t \le \frac{\nu}{\gamma+t}$ where $\nu = \max\left\{\frac{\beta^2(\mathcal{B}+\mathcal{C})}{\beta\mu-1}, (\gamma+1)\Delta_1\right\}$ using an aggregation interval $E \ge 1$ and a diminishing step size $\eta_t = \frac{\beta}{t+\gamma}$ for some $\beta > \frac{1}{\mu}$ and $\gamma > 0$ such that $\eta_1 \le \min\{\frac{1}{\mu}, \frac{1}{4L}\}$ and $\eta_t \le 2\eta_{t+E}$.

First, for $t = 1$ the conclusion holds that $\Delta_1 \le \frac{\nu}{\gamma+1}$ given the conditions. Then by assuming it holds for some $t$, one can derive from (46) that

$$\Delta_{t+1} \le (1 - \eta_t\mu)\Delta_t + \eta_t^2(\mathcal{B} + \mathcal{C})$$

$$\le \left(1 - \frac{\beta\mu}{t+\gamma}\right)\frac{\nu}{\gamma+t} + \left(\frac{\beta}{t+\gamma}\right)^2(\mathcal{B} + \mathcal{C})$$

$$= \frac{t+\gamma-1}{(t+\gamma)^2}\nu + \underbrace{\left[\frac{\beta^2(\mathcal{B}+\mathcal{C})}{(t+\gamma)^2} - \frac{\beta\mu-1}{(t+\gamma)^2}\nu\right]}_{\ge 0}$$

$$\le \frac{t+\gamma-1}{(t+\gamma)^2}\nu$$

$$\le \frac{t+\gamma-1}{(t+\gamma)^2 - 1}\nu$$

$$= \frac{\nu}{t+\gamma+1}, \tag{47}$$

which proves the conclusion $\Delta_t \le \frac{\nu}{\gamma+t}$ for any $t \ge 1$.

Then by the smoothness of the objective function $F$, it follows that

$$\mathrm{E}[F(\bar{w}(t))] - F^* \le \frac{L}{2}\mathrm{E}\|\bar{w}(t) - w^*\|^2$$

$$= \frac{L}{2}\Delta_t \le \frac{L}{2}\frac{\nu}{\gamma+t}. \tag{48}$$

Specifically, by choosing $\beta = \frac{2}{\mu}$ (i.e., $\eta_t = \frac{2}{\mu(\gamma+t)}$), $\gamma = \max\{\frac{8L}{\mu}, E\} - 1$ , we have

$$
\begin{aligned}
\nu &= \max\left\{\frac{\beta^2(\mathcal{B}+\mathcal{C})}{\beta\mu - 1}, (\gamma+1)\Delta_1\right\} \\
&\leq \frac{\beta^2(\mathcal{B}+\mathcal{C})}{\beta\mu - 1} + (\gamma+1)\Delta_1 \\
&= \frac{4(\mathcal{B}+\mathcal{C})}{\mu^2} + (\gamma+1)\Delta_1.
\end{aligned}
\tag{49}
$$

By definition, we have $w(t) = \bar{w}(t)$ at the aggregation steps. Therefore, for $t \in T_A$:

$$
\begin{aligned}
\mathrm{E}[F(w(t))] - F^* &\leq \frac{L}{2}\frac{\nu}{\gamma + t} \\
&= \frac{L}{(\gamma + t)}\left(\frac{2(\mathcal{B}+\mathcal{C})}{\mu^2} + \frac{\gamma+1}{2}\Delta_1\right).
\end{aligned}
$$

$\square$

## D  PROFILE DISSIMILARITY UNDER HOMOMORPHIC ENCRYPTION

The proposed representation profiling scheme encodes the representations of data into a list of distribution parameters, namely $RP(\boldsymbol{w}, D) = \{(\mu_i, \sigma_i^2)|i = 1, 2, \ldots, q\}$ where $q$ is the length of the profile. Theoretically, the information leakage (in terms of the data in $D$) by exposing $RP(\boldsymbol{w}, D)$ is very limited and it is basically impossible to reconstruct the samples in $D$ given $RP(\boldsymbol{w}, D)$. Nonetheless, Homomorphic Encryption (HE) can be applied to the profiles (both locally and on the server) so as to guarantee zero knowledge disclosure while still allowing profile matching under the encryption. In the following we give details on how to encrypt a representation profile and compute profile dissimilarity under Homomorphic Encryption (HE).

To calculate (3) and (4) under encryption, a client needs to encrypt (denoted as $[[\cdot]]$) every single $\mu_i$ and $\sigma_i^2$ in its profile $RP(\boldsymbol{w}, D)$ locally before upload whereas the server does the same for its $RP^*(\boldsymbol{w}, D^*)$. Therefore, according to Eq. (4) we have:

$$
\begin{aligned}
[[\mathrm{KL}(\mathcal{N}_i^{(k)}||\mathcal{N}_i^*)]] =& \frac{1}{2}\log[[(\sigma_i^*)^2]] - \frac{1}{2}\log[[(\sigma_i^{(k)})^2]] - [[\frac{1}{2}]] \\
&+ \frac{([[(\sigma_i^{(k)})^2]] + ([[\mu_i^{(k)}]] - [[\mu_i^*]])^2}{2[[(\sigma_i^*)^2]]},
\end{aligned}
\tag{50}
$$

where the first two terms on the right-hand side require logarithm operation on the ciphertext. However, this may not be very practical because most HE schemes are designed for basic arithmetic operations on the ciphertext. Thus we also consider the situation where HE scheme at hand only provides *additive and multiplicative* homomorphisms (Gentry, 2009). In this case, to avoid the logarithm operation, the client needs to keep every $\sigma_i^2$ in $RP^*(\boldsymbol{w}, D_i)$ as plaintext and only encrypts $\mu_i$, likewise for the server. As a result, the KL divergence can be computed under encryption as:

$$
\begin{aligned}
[[\mathrm{KL}(\mathcal{N}_i^{(k)}||\mathcal{N}_i^*)]] =& \left[\left[\frac{1}{2}\log(\frac{\sigma_i^*}{\sigma_i^{(k)}})^2 + \frac{1}{2}(\frac{\sigma_i^{(k)}}{\sigma_i^*})^2 - \frac{1}{2}\right]\right] \\
&+ \frac{1}{2(\sigma_i^*)^2}([[\mu_i^{(k)}]] - [[\mu_i^*]])^2
\end{aligned}
\tag{51}
$$

where the first term on the right-hand side is encrypted after calculation with plaintext values $(\sigma_i^k)^2$ and $(\sigma^*)^2$ whereas the second term requires multiple operations on the ciphertext values $[[\mu_i^k]]$ and $[[\mu^*]]$.

Now, in either case, we can compute profile dissimilarity under encryption by summing up all the KL divergence values in ciphertext:

$$
[[div(RP_k, RP^*)]] = \frac{1}{q}\sum_{i=1}^{q}[[\mathrm{KL}(\mathcal{N}_i^{(k)}||\mathcal{N}_i^*)]]
$$

$$
\tag{52}
$$

# E    DETAILS OF EXPERIMENTAL SETUP

## E.1    ENVIRONMENT SETUP

Table 4: Experimental setup.

| Setting | Symbol | S-Task | L-Task |
|---|---|---|---|
| Model | $w$ | FFN | CNN |
| Dataset | $D$ | GasTurbine | EMNIST digits |
| Total data size | $\|D\|$ | 36.7k | 280k |
| Validation (benchmark) set size | $\|D^*\|$ | 11.0k | 40k |
| Client population | $N$ | 50 | 1000 |
| Data distribution | - | $\mathcal{N}(514, 154^2)$ | non-IID, dominant$\approx$60% |
| Noise applied | - | fake, gaussian noise | fake, blur, s&p |
| Client specification (GHz) | $s_k$ | $\mathcal{N}(0.5, 0.1^2)$ | $\mathcal{N}(1.0, 0.1^2)$ |
| Comm. bandwidth (MHz) | $bw_k$ | $\mathcal{N}(0.5, 0.1^2)$ | $\mathcal{N}(1.0, 0.1^2)$ |
| Signal-noise ratio | $SNR$ | 1e2 | 1e2 |
| Bits per sample | $BPS$ | 11*8*4 | 28*28*1*8 |
| Cycles per bit | $CPB$ | 300 | 400 |
| # of local epochs | $M$ | 2 | 5 |
| Loss function | $\ell$ | MSE Loss | NLL Loss |
| Learning rate | $\eta$ | 1e-2 | 1e-2 |
| learning rate decay | - | 0.99 | 0.99 |

Detailed experimental settings are listed in Table 4. In the S-Task, the total population is 50 and the data collected by a proportion of the sensors (i.e., end devices of this task) are of low-quality: 10% of the sensors have no valid data and 40% of them produce noisy data. In the L-Task, we set up a relatively large population (1000 end devices) and spread the data (from *EMNIST digits*) across the devices with strong class imbalance – roughly 60% of the samples on each device fall into the same class. Besides, many local datasets are of low-quality: the images on 15% of the clients are irrelevant (valueless for the training of this task), 20% are (Gaussian) blurred, and 25% are affected by the salt-and-pepper noise (random black and white dots on the image, density=0.3). For the S-Task, the maximum number of rounds $t_{max}$ is set to 100 for both aggregation modes, whilst it is set to 300 and 50 for the full aggregation and partial aggregation, respectively, for the L-Task. The preference factor $\alpha$ for our protocol is set to 10. Considering the population of the clients, the setting of the selection fraction $C$ is based on the scale of the training participants suggested by Kairouz et al. (2019).

To simulate a realistic FL system that consists of disparate end devices, the clients are heterogeneous in terms of both performance and communication bandwidth (see Table 4). A validation set is kept by the server (as the benchmark data) and used for model evaluation.

## E.2    DETAILS OF COST FORMULATION

In each FL round, the server selects a fraction (i.e., $C$) of clients, distributes the global model to these clients and waits for them to finish the local training and upload the models. Given a selected set of clients $S$, the time cost and energy cost of a communication round can be formulated as:

$$T_{round} = \max_{k \in S}\{T_k^{comm} + T_k^{train} + T_k^{RP}\}, \tag{53}$$

$$E_k = E_k^{comm} + E_k^{train} + E_k^{RP}. \tag{54}$$

where $T_k^{comm}$ and $T_k^{train}$ are the communication time and local training time, respectively. The device-side energy consumption $E_k$ mainly comes from model transmission (through wireless channels) and local processing (training), corresponding to $E_k^{comm}$ and $E_k^{train}$, respectively. $T_k^{RP}$ and $E_k^{RP}$ estimate the time and energy costs for generating and uploading local profiles and only apply to FEDPROF.

Eq. (53) formulates the length of one communication round of FL, where $T_k^{comm}$ can be modeled by Eq. (55) according to Tran et al. (2019), where $bw_k$ is the downlink bandwidth of device $k$ (in MHz); *SNR* is the Signal-to-Noise Ratio of the communication channel, which is set to be constant as in general the end devices are coordinated by the base stations for balanced *SNR* with the fairness-based policies; $msize$ is the size of the model; the model upload time is twice as much as that for model download since the uplink bandwidth is set as 50% of the downlink bandwidth.

$$
\begin{aligned}
T_k^{comm} &= T_k^{upload} + T_k^{download} \\
&= 2 \times T_k^{download} + T_k^{download} \\
&= 3 \times \frac{msize}{bw_k \cdot \log(1 + SNR)},
\end{aligned}
$$
(55)

$T_k^{train}$ in Eq. (53) can be modeled by Eq. (56), where $s_k$ is the device performance (in GHz) and the numerator computes the total number of processor cycles required for processing $M$ epochs of local training on $D_k$.

$$
T_k^{train} = \frac{M \cdot |D_k| \cdot BPS \cdot CPB}{s_k},
$$
(56)

$T_k^{RP}$ consists of two parts: $T_k^{RPgen}$ for local model evaluation (to generate the profiles of $D_k$) and $T_k^{RPup}$ for uploading the profile. $T_k^{RP}$ can be modeled as:

$$
\begin{aligned}
T_k^{RP} &= T_k^{RPgen} + T_k^{RPup} \\
&= \frac{1}{M}T_k^{train} + \frac{RPsize}{\frac{1}{2}bw_k \cdot \log(1 + SNR)},
\end{aligned}
$$
(57)

where $T_k^{RPgen}$ is estimated as the time cost of one epoch of local training; $T_k^{RPup}$ is computed in a similar way to the calculation of $T_k^{comm}$ in Eq. (55) (where the uplink bandwidth is set as one half of the total $bw_k$); $RPsize$ is the size of a profile, which is equal to $4 \times 2 \times q = 8 \times q$ (four bytes for each floating point number) according to our definition of profile in (2).

Using Eq. (54) we model the energy cost of each end device by mainly considering the energy consumption of the transmitters for communication (Eq. 58) and on-device computation for local training (Eq. 59). For FEDPROF, there is an extra energy cost for generating and uploading profiles (Eq. 60).

$$
E_k^{comm} = P_{trans} \cdot T_k^{comm}
$$
(58)

$$
E_k^{train} = P_f s_k^3 \cdot T_k^{train}
$$
(59)

$$
E_k^{RP} = P_{trans} \cdot T_k^{RPup} + P_f s_k^3 \cdot T_k^{RPgen},
$$
(60)

where $P_f s_k^3$ is a simplified computation power consumption model (Song et al., 2013) and $P_f$ is the power of a baseline processor. $P_{trans}$ is the transmitter's power. We set $P_{trans}$ and $P_f$ to 0.5 and 0.7 Watts respectively based on the benchmarking data provided by Carroll et al. (2010).

