# OpenReview forum: "FedProf: Selective Federated Learning with Representation Profiling"
_ICLR.cc/2022/Conference — ICLR 2022 Submitted_

### Official Review · Reviewer_kUXa · 2021-10-19

**Correctness:** 3
**Technical Novelty And Significance:** 3
**Empirical Novelty And Significance:** 2
**Recommendation:** 5
**Confidence:** 4

**Main Review:**

The key idea of the paper is to select “high-quality” data under the hypothesis noisy and low-quality data has different distributions of representations. This hypothesis needs further justification. First, FL is known for dealing with non-iid heterogeneous datasets. Consider a scenario where clients have high-quality but heterogeneous datasets. How general does this hypothesis hold? In some sense, the idea here is opposite to another idea that has been used for client selection where clients with high loss are more likely to be selected.

Also, I wonder if the algorithm could get stuck in a local suboptimal condition. Specifically, let’s say there are two clusters of clients with different focuses. By chance, initial training selected users from one cluster, then the global model will further select users from the same cluster. Could this happen?

The current evaluation is limited to two simple and homogeneous tasks with added noise, which provides an ideal situation where the proposed algorithm would shine. While these evaluations illustrate the effectiveness of the proposed idea under ideal conditions, more thorough evaluations under general heterogeneous conditions are needed to better understand the pros and cons of the proposed algorithm.

Also, most of the comparison algorithms were proposed in 2019 except one that focuses on communications. Given the fast development of FL, I wonder if there are more recent algorithms that should be compared with.

The authors made an observation that representation layers follow a Gaussian distribution. I understand that this makes it more efficient to calculate distributional difference. However, in principle, this can be applied to cases where Gaussian distribution does not hold, right? Can you elaborate more on this, both in terms of using Gaussian distribution to approximate a non-Gaussian one and using the true non-Gaussian distribution?

In the proofs of Prop. 1 and Prop. 2, central limit theorem is applied and thus requires the Lyapunov’s condition (Def. 1). Could you comment on how restrictive this condition is? For example, in your experiments, does this assumption hold?

One clarification question, in the experiments, how did you select the representation layer?




**Summary Of The Paper:**

In this paper, the authors propose a user selection algorithm for federated learning (FL). The key motivation is to select high quality clients for update and thus to reduce the impact of low quality data on FL training. A hidden hypothesis is that high quality data has similar representations while noisy and low-quality data has different distributions of representations. Based on this hypothesis, the key idea is to select users based on their representation layer distribution difference from the global model, the higher the difference, the lower the chance of the client being selected.  Furthermore, the authors observed and proved that representation layers follow a Gaussian distribution, which makes the representation difference learning more efficient. The authors evaluated the proposed algorithm in comparison to existing algorithms using a small-scale sensor dataset and a large-scale EMNIST dataset. The proposed algorithm performs well in the evaluation.

**Summary Of The Review:**

In summary, the authors propose to more favorably select FL clients whose representation distribution is more similar to that of the global model. While the current evaluation results are encouraging, data heterogeneity in FL and model robustness need to be carefully considered.

---

> ### Author Response · Authors · 2021-11-18
> **Thanks for your review. Our responses are as the following.**
>
> - We believe the hypothesis holds for noisy and low-quality data because they change the marginal distribution $P(X)$ in the feature space and consequently it leads to a drift of the posterior $P(H|X)$ for any layer of representations $H$. Our empirical observations (Fig. 6) also reflect the fact that data quality has a strong correlation with representation distribution and thus affects the behavior of FedProf. On the other hand, heterogeneous datasets (in terms of class distributions) do not necessarily display significant distributional differences -- our **L-task** is set up with a multitude of heterogeneous local datasets where the class distributions are very different (see Sec 5.1) from each other and the benchmark dataset.
> - We have seen opposite ideas behind the studies on loss-based client selection. Some prioritize high-loss clients (Goetz et al. 2019) whilst some filter out high-loss local data to improve convergence (Tuor et al. 2021). We believe that representations are more informative than loss because both hard samples and abnormal ones yield high values of loss.
> - It is possible that FedProf could lead to suboptimal global model by overfitting a certain group of the clients. But this can be avoided in two ways. First, just like any algorithms that introduce preference into the policy, our method has a tunable preference factor $\alpha$ that controls its focus. With a smaller $\alpha$, our algorithm behaves more randomly and thus can avoid overfitting ($\alpha=0$ degenerates it to FedAvg). Second, FL was designed for large-scale learning scenarios where data redundancy can often be expected. In this case, a small group of users with quality data may well suffice for training a decent global model (as revealed in Fig. 6).
> - We do have experiments over both homogeneous data (S-Task) and **heterogenous data settings (L-Task)**. In S-Task, data are spread unevenly with concept drift. In L-Task, local data are heterogenous in class distribution and also class-imbalanced. It was done by making clients have different dominant classes (i.e., a class label that accounts for roughly 60% of local data).
> - Most recent studies that utilize representation information place focus on objective adaptation or model distillation. We think our algorithm should not be put in the same group. The choices of our baselines are mainly related to client selection/sampling based FL algorithms. We will consider additional comparison in the revision.
> - It is an interesting point about how to compare representations that may not follow Gaussian. So far we haven't seen any distributional approximation approaches other than ours. Related studies often calculate the similarity/distance between representations directly as vectors (Li et al. 2021; Chen et al. 2020). For approximating non-Gaussian representation patterns (if any), vector similarity/distance could be the only reasonable method. So far we haven't found other theoretical guarantees from the distributional perspective other than our propositions.
> - The Lyapunov's condition basically states that the overall variation of a set of random variables is bounded. With the increase of dimensionality, the condition gets more tolerant to extreme values. For neural nets, extreme variations are rare to see and usually means bad practice. Practically we found the condition well satisfied when the global model is properly initialized and batch normalization is applied.
> - We would like to explain that we selected the first hidden layer for representation extraction. There are two reasons behind it. First, we want to keep as much low-level representation information as possible. Second, layers close to the output are more susceptible to data reconstruction once the representations are disclosed. In practice, our method can use any or any combinations of layers if the situation permits.
>
> [1] Goetz, J., Malik, K., Bui, D., Moon, S., Liu, H., & Kumar, A. (2019). Active federated learning. arXiv preprint arXiv:1909.12641.
>
> [2] Tuor, T., Wang, S., Ko, B. J., Liu, C., & Leung, K. K. (2021, January). Overcoming noisy and irrelevant data in federated learning. In 2020 25th International Conference on Pattern Recognition (ICPR) (pp. 5020-5027). IEEE.
>
> [3] Li, Q., He, B., & Song, D. (2021). Model-Contrastive Federated Learning. In Proceedings of the IEEE/CVF Conference on Computer Vision and Pattern Recognition (pp. 10713-10722).
>
> [4] Chen, T., Kornblith, S., Norouzi, M., & Hinton, G. (2020, November). A simple framework for contrastive learning of visual representations. In International conference on machine learning (pp. 1597-1607). PMLR.

---

> > ### Comment · Reviewer_kUXa · 2021-11-19
> > **Representation shift clarification**
> >
> > Thank you for your responses. Could you elaborate on the following points?
> >
> > "the hypothesis holds for noisy and low-quality data because they change the marginal distribution  P(X) in the feature space."
> >
> > Could you elaborate more on this? Also, how significant does the noise or distortion needed to be?
> >
> > "heterogeneous datasets (in terms of class distributions) do not necessarily display significant distributional differences "
> >
> > As you stated, this is "not necessarily", meaning it could happen sometimes. Could you eleborate more on this? I understand your example. However, one example is not enough to address this question.
> >
> > Thank you.

---

> > > ### Author Response · Authors · 2021-11-19
> > > **Regarding representation shift**
> > >
> > > Sure.
> > > The hypothesis ("noisy and low-quality data has different distributions of representations") can be explained in the following way. Imagine that we are working on a 1-D feature space $\chi$ where the samples' feature $X$ follows some distribution (can be any distribution) $P(X)$. Now let's see what happens when noises are applied and when data are heterogeneous, respectively.
> > > - With noises blended into the data, the distribution of samples gets distorted, which means a change to the marginal distribution $P(X)$. For example, if the user device introduces some noise to the data (e.g., white noise, blur, low-pass filters, etc.), the resulting data $X+noise$ could follow any distribution $P(X+noise)$ that is completely different from the original $P(X)$. How significant the distortion is depends on the noise intensity. For example, we observed that Gaussian blur has stronger influence on the representation distributions than the Salt&Pepper noise we applied to the images (see Fig. 6(b) in the revised paper).
> > > - By contrast, heterogeneous local data means non-uniform sampling from the population distribution $P(X)$. It means that $P(X)$ is never distorted but the clients are sampling from it in different manners. It is possible to see "significant distributional differences" if extremely biased sampling is performed in the feature space. But we have observed clients with totally different label distribution $P(Y)$ and potentially different posterior $P(Y|X)$ producing similar representation distributions given the same global model. This is supported by our results shown in Fig. 6 where "clients with normal data" are totally heterogeneous.
> > >
> > > Thank you.

---

> > > > ### Comment · Reviewer_kUXa · 2021-11-29
> > > > **Thank you**
> > > >
> > > > Thank you for your clarification.

---

### Official Review · Reviewer_5Q6Y · 2021-11-01

**Correctness:** 3
**Technical Novelty And Significance:** 2
**Empirical Novelty And Significance:** 3
**Recommendation:** 6
**Confidence:** 3

**Main Review:**

Strengths: The proposed method is intuitive and simple to implement. It also comes with theoretical guarantees (though the assumptions are highly idealized).

Weaknesses and some questions:
- If I understand correctly (pls correct me if I'm wrong), Proposition 1 and 2 are consequences of Lyapunov CLT, thus one needs each coordinate of the representation vector to be independent. It is not clear to me if the independence assumption would hold in practice.
- Following my previous point, I am curious about what will happen if we replace the product Gaussian (Eq. 2) by a Gaussian with some non-identity covariance matrix.
- Is the choice of KL divergence (Eq. 4) essential for the performance? What will happen if we choose other metrics?
- How sensitive is the performance with respect to the choice of $\alpha$?

**Summary Of The Paper:**

This paper proposes FedProf, a new client sampling scheme that speeds up the convergence of FedAvg type algorithms. The author proves convergence of FedProf under a set of simplifying conditions and they demonstrate the utility of FedProf via empirical studies.

**Summary Of The Review:**

I appreciate the simplicity of the idea and the performance boost it brings about. From what I understand, it can be applied to any FL algorithms that involve client sampling. Therefore, I think this paper deserves 6: marginally above the acceptance threshold.

---

> ### Author Response · Authors · 2021-11-18
> **Thank you for the comments and we have the following responses.**
>
> - It is true that CLT and Lyapunov's CLT make independence assumption on the random variables and we can prove that it holds for hidden layers' outputs. We explain it with Bayesian network concepts. Assume we have two hidden neurons $i$ and $j$ that can be abstracted as two functions $f_i$ and $f_j$. Appparently with the input variable $X$, their outputs are two random variables $H_i$ and $H_j$. Because $H_i=f_i(X)$ (same for $H_j$), we have dependencies $X$->$H_i$ and $X$->$H_j$. According to Local Markov Property (each variable is conditionally independent of its non-descendants given its parent), $H_i$ is independent of $H_j$ given the value of $X$. We have added a brief explanation to our revised paper to address this concern.
> - Based on my understanding (pls correct me if wrong), using a non-identity covariance matrix makes the profile a multivariate Gaussian, which is also feasible for in-memory storage. But so far we cannot find any way to efficiently calculate the distance between two multivariate Gaussian distributions. Though our component-wise representation profiles look simple, but it allows for comparing any two profiles very very efficiently via Eq. (4) where no summation or integral is needed.
> - We thought about distributional distances including KL, Hellinger, KS test, etc. The problem is that: all these distances involve discrete summation or continuous integral over the PDFs or CDFs and may also involve sampling (which complicates the profiling with more parameters). So the use of KL is kind of essential as we can leverage the derived formula Eq. (4) by Roberts & Penny (2002). On the one hand, we can compare two profiles in $O(1)$. On the other hand, Homomorphic Encryption (HE) can be applied for privacy protection (Appendix D).
> - Through empirical study we found that our algorithm is not very sensitive when $\alpha$ is set within [5,30]. With $\alpha$ approaches 1, our algorithm degenarates to FedAvg. With $\alpha$ larger than 30, the algorithm has tendency to overfit some of the local datasets and in this case, the performance largely depends on the how similar they are to the test set. It takes a bulk of content for parameter study, we will consider discussing it in a new appendix if possible.

---

> > ### Comment · Reviewer_5Q6Y · 2021-11-26
> > **Reply to authors' response**
> >
> > I thank the authors for the detailed response. Most of my concerns are properly addressed, except for the one regarding Prop. 1 and 2. I agree with Reviewer pcRb that "in Proposition 1 and 2, all non-linear components in the neural network seem to be bypassed and the statements are simply equivalent ways of stating CLT".

---

> > > ### Author Response · Authors · 2021-11-29
> > > **Further response**
> > >
> > > Thank you. We think what we are doing to deal with the representations from non-linear operators is essentially studying the **joint distribution** of them in the form of their linear combinations. 'Bypass' doesn't sound very precise to us because we never change any transformation during the forward or backward propagation. We make the distributional analysis possible for non-linear operators with the fusion trick.

---

### Official Review · Reviewer_1UFK · 2021-11-02

**Correctness:** 4
**Technical Novelty And Significance:** 2
**Empirical Novelty And Significance:** 2
**Recommendation:** 5
**Confidence:** 4

**Main Review:**

This paper is well-written and easy to understand. The motivation is clear and intuitive. The proposed method is technically sound and the results are strong.

I have a few comments, particularly for the evaluation part.


- More Datasets: it seems to work well on the reported datasets. I wonder if the tendency is still consistent with more benchmark datasets. I would like to suggest common datasets such as CIFAR-10 and CIFAR-100. I also would like to see the results when the number of classes is increased.

- Standard Deviation: Can you provide statistical information in Table 2 and 3? I would like to see the variance of the proposed model. Few performances seem marginal compared to the base models.


- The Current Baseline Models: I found that the baseline models used in the paper are slightly outdated. I suggest [1], the state-of-the-art method tackling similar scenarios, via performing model-level contrastive learning.


- Overview: Please consider adding the concept illustration for FedProf, which can be helpful for common readers to capture high-level ideas.

[1] Model-Contrastive Federated Learning, CVPR'21





**Summary Of The Paper:**

The authors tackle the federated learning scenarios where local data is biased, noisy, or even irreverent, which could significantly degenerate the global performance, and the authors propose FedProf, which utilizes data representation profiling and matching to mitigate the impact of low-quality clients during training.


**Summary Of The Review:**

Overall, I enjoyed reading the paper, and I'll raise my score if my concerns are properly addressed.

---

> ### Author Response · Authors · 2021-11-17
> **Thanks a lot for you comments.**
>
> 1. Actually we did have experiments on CIFAR-100 and we show some intermediate results in Fig. 2(a) and (b) in the paper. The reasons why we display GasTurbine and EMNIST in the evaluation section are: i) GarTurbine is a regression task by which we can justify if our method works on shallow MLP models, and ii) EMNIST is easier than CIFAR-100 but is large enough to support a large-scale client distribution. With CIFAR-10/100, the common scale are 10~100 devices; but with EMNIST we scaled out to 1000. We believe it is important for FL algorithms to show effectiveness under large-scale settings. We will consider enriching our numerical results in the revision.
> 2. We have added standard deviations to the stats in Tables 2 and 3 for our algorithm as adviced. Though AFL attained close results in terms of rounds/time under partial aggregation, our algorithm has a clear edge in the best accuracy obtained.
> 3. MOON (Model-Contrastive Federated Learning) is a good baseline for tackling similar scenarios. However, it is only applicable to Cross Entropy Loss for classification. So unfortunately MOON does not work for our S-task. We did include it in our related work and have extended its discussion considering its potential as a baseline.
> 4. We have added a workflow overview (see Fig. 3 in the revised version) as suggested to illustrate our framework for better understanding of the ideas behind our work.

---

> > ### Comment · Reviewer_1UFK · 2021-12-01
> > **Thank you**
> >
> > Thank the authors for your clarification and updating the new revision.

---

### Official Review · Reviewer_pcRb · 2021-11-04

**Correctness:** 3
**Technical Novelty And Significance:** 2
**Empirical Novelty And Significance:** 3
**Recommendation:** 3
**Confidence:** 3

**Main Review:**

Strenth:

I personally like the style of this work, which attempts to formulate and present results on FL in a more theoretical way.

Weakness:

Overall the presentation on theoretical results can be improved, and the connection of some statements to the main results is not clear.

1. For example in Proposition 1 and 2, all non-linear components in the neural network seem to be bypassed and the statements are simply equivalent ways of stating CLT. If it is the case, this needs to be clearly mentioned before those propositions. Besides, it is also not clear to me that how this over-simplified model is related to the rest of the works.

2. The main Theorem needs to be better stated and explained. For example, the most relevant question to the readers is how the construction of profiles affects the performance of the algorithm.

3. To better understand the main Theorem, it would help if the stated learning bound can be compared to some benchmarks, e.g., the performances of using fixed \lambda.

4. Minor: the last equations in all Proposition proofs are stated in terms of convergence in distribution, which may not hold.



**Summary Of The Paper:**

This paper proposes a federated learning algorithm called FedProf, by using a training rule that updates the model based on divergences between data representation.

**Summary Of The Review:**

Overall, the results in this paper can be better presented, and the connections between certain statements are not clear. Therefore, I recommend a rejection.

---

> ### Author Response · Authors · 2021-11-17
> **Thank you for your approval and the comments provided.**
>
> 1. We are not actually bypassing the non-linear components but extracting representations before activation (Proposition 1) or after a simple additional fuse operation (Proposition 2). The propositions justify our representation profiling and matching method which essentially hypothesizes that representations can be compressed and compared in the form of Gaussian distributions. We will clarify this in the revision.
> 2. The construction of representation profiles provides an efficient way to compare different local datasets' value. On this basis, our algorithm provides a heuristic approach to evaluating user models importance by comparing representations so as to optimize our objective function inspired by Agnostic Federated Learning (Mohri et al. ICML'19) where the best weights of users' models are unknown. Due to multiple sources of randomness (local SGD and client selection), we present the main Theorem to prove that, under partial participation and aggregation and this policy, the regret is upper bounded in expectation.
> 3. With fixed $\lambda$ (which means the selection probability distribution does not change), we believe the optimality of the global model depends on the relation between the optimal weights $\rho$'s and the selection probability distribution $\lambda$'s, which makes three difference cases. In case 1, $\rho$'s match $\lambda$'s precisely and our Theorem holds. In case 2, $\lambda$'s are equally set as $1/n$ and our algorithm degenerates to FedAvg whose convergence upper bound has been extensively studied. In case 3, $\lambda$'s are set randomly, which cannot yield any performace guarantee.
> 4. Convergence in distribution is what Lyapunov's CLT can guarantee for the distribution of latent representations as random variables. Theoretically our result holds but practically how well the pattern matches the result mainly depends on the dimensionality of the representations to profile. High-dimensional representations tend to show clearer and stabler Gaussian patterns (Fig. 2).

---

> > ### Comment · Reviewer_pcRb · 2021-11-24
> > **Response**
> >
> > 1. In proposition 2, it is assumed the features are Lyapunov after non-linear operators. This is the bypassing I was referring to because the assumption is known to be not true in general. Otherwise, it is impossible for GAN to function.
> >
> > 2. Given the current presentation, the reader would be interested in concrete examples and what each variable means. E.g., what happens when the profiles are irrelevant to the data but their KL divergences are small. I would expect the RHS of the inequality to be large but which terms reflect this phenomenon. All intermediate variables being introduced in the theorem should either be explained in this way or merged into other variables, otherwise, it is difficult for the readers to appreciate the value of the proposed learning bound.
> >
> > 4. Please note that the convergence in distribution guaranteed by Lyapunov's CLT is exactly the equations before the ones mentioned in my earlier comment. This is a terminology issue but such arguments often create confusion to the reader as they contradict other elementary facts.

---

> > > ### Author Response · Authors · 2021-11-29
> > > **Further response**
> > >
> > > Thanks for the response.
> > > 1. It is a good point to bring **Generative Models** into the discussion. When it comes to our Proposition, we actually got some inspiration from Variational AutoEncoders (VAEs) which, in a sense, works in very similar way to GANs. Multiple non-linear operators can be applied to both VAEs and GANs for generating more diverse features. The most important assumption for VAEs is that the output features all follow Gaussian, which automatically proves our Proposition even without the Lyapunov's condition (because the sum of Gaussians follows a Gaussian). GANs may not need the same assumption, but we don't think non-linear operatiors will certainly create significant feature variations (that break the Lyapunov's condition) when looking at the whole latent space. Moreover, as we state in the paper, the condition can be ensured with **Batch Normalization**.
> > > 2. For the convergence guarantee, it is provided on the basis of the hypothesis that the (dis)similarity between representation profiles is determined by the (dis)similarity between data distributions. Without making any assumption on the data, the Theorem is providing a theoretical upper bound given the optimal preferences (i.e., an oracle case). The key variables (e.g., learning rate, intervals, $L$-smoothness, $\mu$-convexity, etc.) in the Theorem have been explained in or prior to the theorem body whilst intermediate variables (e.g., $B$, $C$, $\Gamma$, etc.) are mainly used for making the RHS concise. We have explained them in the theorem as well as in the appendix.
> > > 3. Thanks for the advice and we will clarify them.

---

### Decision · Program_Chairs · 2022-01-20

**Decision:**

Reject

**Comment:**

This paper proposes a federated learning method called FedProf that adaptively selects different subsets of the clients' data in training the global model. There were several concerns brought up in the reviews and discussion. The multivariate Gaussian (with identity covariance) assumption on the neural network representation is limited. The paper also claimed to provide privacy preservation, but there is no formal statement of the actual privacy guarantees. (The fact that it's running federated learning does not guarantee privacy protection.) The presentation could use improvement. The reviewers had issues trying to understand the main theorem. Overall, there is not sufficient support for acceptance.